# TriRE: A Multi-Mechanism Learning Paradigm for Continual Knowledge Retention and Promotion

**Preetha Vijayan[1],*, Prashant Bhat[2,3,*], Bahram Zonooz[2,3,†], Elahe Arani[2,†]**

[1]NavInfo Europe    [2]Eindhoven University of Technology (TU/e)    [3]TomTom

preetha.vijayan@navinfo.eu, {p.s.bhat, e.arani, b.zonooz}@tue.nl

## Abstract

Continual learning (CL) has remained a persistent challenge for deep neural networks due to catastrophic forgetting (CF) of previously learned tasks. Several techniques such as weight regularization, experience rehearsal, and parameter isolation have been proposed to alleviate CF. Despite their relative success, these research directions have predominantly remained orthogonal and suffer from several shortcomings, while missing out on the advantages of competing strategies. On the contrary, the brain continually learns, accommodates, and transfers knowledge across tasks by simultaneously leveraging several neurophysiological processes, including neurogenesis, active forgetting, neuromodulation, metaplasticity, experience rehearsal, and context-dependent gating, rarely resulting in CF. Inspired by how the brain exploits multiple mechanisms concurrently, we propose TriRE, a novel CL paradigm that encompasses *retaining* the most prominent neurons for each task, *revising* and solidifying the extracted knowledge of current and past tasks, and actively promoting less active neurons for subsequent tasks through *rewinding* and relearning. Across CL settings, TriRE significantly reduces task interference and surpasses different CL approaches considered in isolation.[1]

## 1 Introduction

Continual learning (CL) over a sequence of tasks remains an uphill task for deep neural networks (DNNs) due to catastrophic forgetting of older tasks, often resulting in a rapid decline in performance and, in the worst-case scenario, complete loss of previously learned information [38]. Several approaches, such as parameter isolation [43, 4], weight regularization [56, 45], and experience rehearsal [41, 42, 9] have been proposed in the literature to address the problem of catastrophic forgetting in DNNs. Despite their relative success, these research directions have predominantly remained orthogonal and suffer from several shortcomings. Parameter isolation approaches suffer from capacity saturation and scalability issues in longer task sequences, while weight regularization approaches cannot discriminate classes from different tasks, thus failing miserably in scenarios such as class-incremental learning (Class-IL) [28]. In scenarios where buffer size is limited due to memory constraints (e.g., edge devices), rehearsal-based approaches are prone to overfitting on the buffered data [7]. As these research directions have rarely crossed paths, there is a need for an integrated approach to leverage the advantages of competing methods to effectively mitigate catastrophic forgetting in CL.

Catastrophic forgetting is a direct consequence of a more general problem, namely the stability-plasticity dilemma [2, 36]: the extent to which the CL model must be plastic to accommodate newly acquired knowledge and stable to not interfere with previously learned information [38]. In stark

---

*Equal contribution.    †Equal advisory role.

[1]Code is available at `https://github.com/NeurAI-Lab/TriRE`

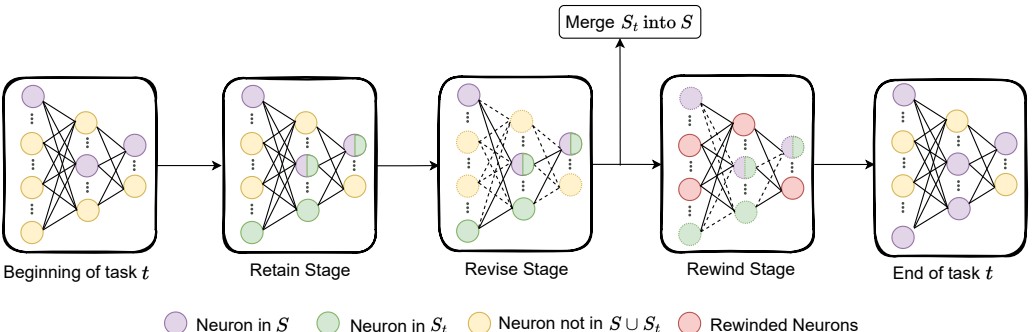

Figure 1: *TriRE* consists of a three-phase learning paradigm that reduces task interference and drastic weight changes by using task modularity. In the *Retain* stage, the method selects and preserves the most active neurons and weights in a mask $\mathcal{S}_t$, which is used in the subsequent *Revise* stage to finetune the joint distribution of current and past tasks along with a cumulative subnetwork mask $\mathcal{S}$. The *Rewind* stage is responsible for reintroducing less active neurons to the learning process for future tasks by actively forgetting and relearning the non-sparse subnetwork.

contrast to DNNs, biological systems manage this dilemma better and are able to learn continually throughout their lifetime with minimal interference. CL in the brain is administered by a rich set of neurophysiological processes that encompass different kinds of knowledge, and conscious processing that integrates them coherently [17]. Empirical studies suggest that metaplasticity [27] and experience replay play a prominent role in memory consolidation in the brain [40, 15]. In addition, neurogenesis in the brain is crucial for the growth and restructuring necessary to accommodate new skills [3]. Neuromodulatory systems facilitate swift learning and adaptability in response to contextual changes induced by new stimuli or shifts in motivation [33]. Whereas context-dependent gating [34] and active forgetting [19] improve the separation between the representations of patterns belonging to different tasks. By simultaneously leveraging these processes, the brain exploits task similarity and exhibits positive forward transfer, rarely resulting in catastrophic forgetting.

Inspired by the biological underpinnings of the CL mechanisms in the brain, we propose *'REtain, REvise & REwind' (TriRE)*, a novel CL paradigm to mitigate catastrophic forgetting. Specifically, TriRE involves experience rehearsal, scalable neurogenesis, selective forgetting, and relearning to effectively mitigate catastrophic forgetting. Within each task, the proposed method consists of three stages: (i) *Retain*, where the most active neurons and their corresponding most important weights of the task are extracted and retained to avoid task interference and drastic weight changes, (ii) *Revise*, where the extracted network is finetuned to revise and solidify the current task as well as the joint distribution of the past tasks, (iii) *Rewind*, where the free neurons undergo active forgetting and relearning to promote the less active neurons back into the learning circuit for the next task as illustrated in Figure 1. TriRE effectively combines multiple mechanisms and leverages the advantages offered by different CL approaches.

We find that TriRE significantly reduces task interference and surpasses the aforementioned CL approaches across various CL scenarios. Specifically, TriRE outperforms rehearsal-based approaches in Seq-TinyImageNet for Class-IL scenario by almost 14%, even under low-buffer regimes, by promoting generalization through weight and function space regularization. Experience rehearsal enables discrimination of classes belonging to different tasks in TriRE, resulting in at least twice as good performance over weight regularization methods for the same dataset. Unlike parameter isolation approaches, TriRE is scalable and produces at least a 7% relative improvement compared to parameter isolation approaches in Seq-CIFAR100 Task-IL setting without requiring access to task identity at inference time.

## 2   Related works

**Rehearsal-based Approaches:** Prior works attempted to address the problem of catastrophic forgetting by explicitly storing and replaying previous task samples, akin to experience rehearsal in the brain. Experience rehearsal (ER) approaches [41, 42] maintain a fixed capacity memory buffer to

store data sampled from previous task distributions. Several approaches are built on top of ER to better preserve the previous task information: GCR [49] proposed a coreset selection mechanism that approximates the gradients of the data seen so far to select and update the buffer. DER++ [9] and CLS-ER [6] enforce consistency in predictions using soft targets in addition to ground-truth labels. DRI [52] uses a generative model to further support experience rehearsal in low buffer regimes. More recent works like TARC [8], ER-ACE [11] and Co$^2$L [12] focus on reducing representation drift right after task switch to mitigate forgetting through asymmetric update rules. Under low-buffer regimes and longer task sequences, however, these approaches suffer from overfitting on the buffered samples.

**Weight Regularization Approaches:** Catastrophic forgetting mainly emanates from large weight changes in DNNs when learning a new task. Therefore, weight-regularization methods seek to penalize the sudden changes to model parameters that are crucial for the previous tasks. Depending on the type of regularization, these approaches can be broadly categorized into prior- and data-focused approaches. Prior-focused approaches, such as elastic weight consolidation (EWC) [25], online EWC (oEWC) [45], memory-aware synapses (MAS) [5], and synaptic intelligence (SI) [56] employ prior information on model parameters and estimate the importance of parameters associated with previous tasks based either on the gradients of the learned function output or through Fisher's information matrix. On the other hand, data-focused methods, such as Learning without Forgetting (LwF) [30] instead perform knowledge distillation from models trained on previous tasks when learning on new data. Although weight regularization approaches do not require a memory buffer and are scalable, they only impose a soft penalty, thus failing to entirely prevent forgetting of previous task information.

**Parameter Isolation Approaches:** Parameter isolation has been predominantly done in two ways: either within a fixed capacity or by growing in size. In the former, dynamic sparse methods such as PackNet [32], CLNP [16], PAE [23], and NISPA [18] make use of DNN's over-parameterization to learn multiple tasks within a fixed model capacity. Similar to the brain, these models simultaneously learn both connection strengths and a sparse architecture for each task, thereby isolating the task-specific parameters. However, these methods suffer from capacity saturation in longer task sequences, limiting their ability to accommodate new tasks. In contrast, the latter methods, such as PNNs [39], Expert-gate [4] and DEN [55] expand in size, either naively or intelligently, to accommodate new tasks while minimizing forgetting. Although these approaches are extremely efficient in mitigating catastrophic forgetting, they do not scale well with longer task sequences, rendering them inapplicable in real-world scenarios.

Contrary to DNNs, the brain simultaneously exploits multiple neurophysiological processes, including neurogenesis [3], active forgetting [19], metaplasticity [27], experience rehearsal [40], and context-dependent gating [34] to continually acquire, assimilate, and transfer knowledge across tasks without catastrophic forgetting [26]. Inspired by how the brain exploits multiple mechanisms concurrently, we propose a novel CL paradigm, TriRE, that leverages the advantages of multiple aforementioned mechanisms to effectively mitigate catastrophic forgetting in CL.

## 3 Method

CL problems typically comprise $t \in \{1, 2, .., T\}$ sequential tasks, with $c$ classes per task, and data that appear gradually over time. Each task has a task-specific data distribution associated with it $(x_t, y_t) \in D_t$. We take into account two well-known CL scenarios, class-incremental learning (Class-IL) and task-incremental learning (Task-IL). Our working model consists of a feature extractor network $f_\theta$ and a single head classifier $g_\theta$ that represents all classes of all tasks.

Sequential learning through DNNs has remained a challenging endeavor, since learning new information tends to dramatically degrade performance on previously learned tasks. As a result, to better retain information from past tasks, we maintain a memory buffer $D_m$ that contains data from tasks previously viewed. Considering the desiderata of CL, we assume that the model does not have infinite storage for previous experience and thus $|D_m| \ll |D_t|$. To this end, we use *loss-aware balanced reservoir sampling* [10] to maintain the memory buffer. We update the working model, $\Phi_\theta = g_\theta(f_\theta(.))$, using experience rehearsal at each iteration by sampling a mini-batch from both $D_t$ and $D_m$ as follows:

$$\mathcal{L} = \underbrace{\mathbb{E}_{(x_i, y_i) \sim D_t} [\mathcal{L}_{ce}(\sigma(\Phi_\theta(x_i)), y_i)]}_{\mathcal{L}_t} + \lambda \underbrace{\mathbb{E}_{(x_j, y_j) \sim D_m} [\mathcal{L}_{ce}(\sigma(\Phi_\theta(x_j)), y_j)]}_{\mathcal{L}_{er}}, \qquad (1)$$

where $\mathcal{L}_{ce}$ is the cross-entropy loss, $\sigma(.)$ is the softmax function, $\mathcal{L}_t$ is the task-wise loss and $\mathcal{L}_{er}$ is the rehearsal-based loss. The objective in Eq. 1 encourages plasticity through the supervisory signal from $D_t$ and increases stability through $D_m$. However, as CL training advances, model predictions carry more information per training sample than ground truths [7]. Hence, soft targets can be utilized in addition to ground-truth labels to better preserve the knowledge of the earlier tasks. Traditional methods to enforce consistency in predictions include using an exponential moving average (EMA) of the weights of the working model [6] or holding previous predictions in a buffer [9]. As the former result in better knowledge consolidation and decision boundaries, we use EMA of the working model weights to ensure consistency in predictions:

$$\mathcal{L}_{cr} \triangleq \mathop{\mathbb{E}}_{(x_j, y_j) \sim D_m} \|\Phi_{\theta_{EMA}}(x_j) - \Phi_\theta(x_j)\|_F^2, \tag{2}$$

where $\Phi_{\theta_{EMA}}$ is the EMA of the working model $\Phi_\theta$ and $\| \, . \, \|_F$ is the Frobenius norm. We update the EMA model as follows:

$$\theta_{EMA} = \begin{cases} \mu \, \theta_{EMA} + (1 - \mu) \, \theta, & \text{if } \zeta \geq \mathcal{U}(0,1) \\ \theta_{EMA}, & \text{otherwise} \end{cases} \tag{3}$$

where $\mu$ is the decay parameter and $\zeta$ is the update rate. Finally, the EMA model acts as a self-ensemble of models with distinct task specializations for inference rather than the working model.

**Notations:** Let $\mathcal{S}_t$ be the extracted subnetwork mask corresponding exclusively to the current task and $\mathcal{S}$ be the cumulative dynamic network mask corresponding to the tasks learned so far. At the end of the training, $\mathcal{S}$ would contain the most active neurons and the best corresponding weights across all tasks. In the following, we describe in detail various components of TriRE learning paradigm.

## 3.1 Retain

In CL, typically, models from previous tasks are seen as initialization and are "washed out" by new updates from the current task, which causes CF [35]. However, the brain uses context-dependent gating [26] to selectively filter neural information based on the context in which it is presented, allowing the development of specialized modules that can be added or removed from the network without disrupting previously learned skills [37, 50]. Inspired by this, the *Retain* phase induces modularity in the model by training a hyper-network first and then extracting a subnetwork that is equivalently representational of the current task knowledge. This extracted subnetwork not only helps in creating task-wise specialized modules, but also helps the model preserve capacity for future tasks. Retention of this subnetwork is done using heterogeneous dropout of activations and weight pruning.

The *Retain* stage appears at the beginning of each task. At this stage, initially $\{f_\theta \mid \theta \notin \mathcal{S}\}$ is trained using a mini-batch of $D_t$ and $\{f_\theta \mid \theta \in \mathcal{S}\}$ is trained using a mini-batch of $D_m$. This is to ensure that the weights not in the cumulative network learn the new task to maintain plasticity, while the weights in the cumulative network learn a combination of old and new tasks to maintain stability. At the convergence of this training, we perform activation pruning followed by weight pruning to extract $\mathcal{S}_t$ as shown in Figure 2.

**Activation Pruning:** This involves identifying and extracting neurons that contribute the most to the overall activation or output of the network. We monitor the frequency of neuron activations when a network is trained on a task. In essence, each neuron is given an activation counter that increases when a neuron's activation is among the top-k activations in its layer. We use these activation counts to extract the k-winner activations and retain them as the knowledge base for the current task. Heterogeneous dropout [1] is used to map the activation counts of each neuron to a Bernoulli variable, indicating whether the said neuron is extracted or dropped. This, essentially, leaves the less activated neurons free to learn the next tasks.

**Weight Pruning:** After retaining the most activated neurons for the task, we prune the less important connections corresponding to these neurons. In contrast to conventional methods, which only leverage weight magnitude or Fisher information for pruning, our method also takes into account the significance of weights with respect to data saved in the rehearsal buffer. Continual Weight Importance (CWI) [53] criteria ensure that we maintain: (1) weights of greater magnitude for output stability, (2) weights significant for the current task for learning capacity, and (3) weights significant for past data to prevent catastrophic forgetting.

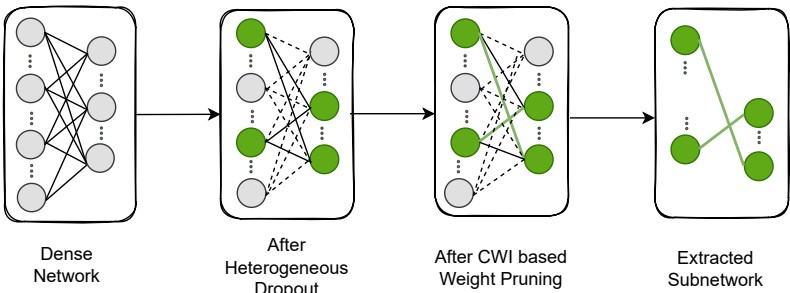

Figure 2: Schematic representation of extraction of subnetwork at the end of *Retain* stage. The dense network is first pruned using k-WTA criteria, resulting in a subnetwork of the most activated neurons. This subnetwork is then pruned using CWI criteria, resulting in a final extracted subnetwork, $\mathcal{S}_t$.

For the working model, the CWI of weight $\theta$ is defined as follows,

$$CWI(\theta) = \|\theta\|_1 + \alpha\|\frac{\delta\tilde{\mathcal{L}}_{ce}(D_t;\theta)}{\delta\theta}\|_1 + \beta\|\frac{\delta\mathcal{L}_{ce}(D_m;\theta)}{\delta\theta}\|_1 \tag{4}$$

where $\alpha$ and $\beta$, are coefficients that regulate the weight of current and buffered data, respectively. In addition, $\tilde{\mathcal{L}}_{ce}$ denotes the single-head form of the cross-entropy loss, which only takes into account the classes relevant to the current task by masking out the logits of other classes.

At the end of this stage, we end up with a mask of the most activated neurons and their most important weights for the current task, $\mathcal{S}_t$.

## 3.2 Revise

Although context-dependent gating is a helpful phenomenon for CL in tackling CF, there are other biological phenomena such as neuromodulation, metaplasticity, and neurogenesis among others which also contribute to the brain's ability to combat CF [26]. Neuromodulation, for instance, is used to decide whether a new feature is novel and unfamiliar (that is, creates a new memory) or common and familiar (that is, consolidates into an existing memory). Taking the cue from that, *Revise* stage mainly focuses on revising and solidifying the knowledge that the extracted subnetwork of the current task, $\mathcal{S}_t$, and the cumulative subnetwork of past tasks, $\mathcal{S}$, currently possess. It is clear that $\mathcal{S}_t$ is specialized on the current task and $\mathcal{S}$ is specialized on the past tasks. However, finetuning these two networks jointly can improve the knowledge overlap, and the features thus generated would be a better approximation of all the seen classes.

Firstly, $\{f_\theta \mid \theta \notin (\mathcal{S} \cap \mathcal{S}_t)\}$ is trained with a mini-batch of $D_t$ so that a part of the extracted network, $\mathcal{S}_t$, can be utilized to regain the performance for the current task. Subsequently, $\{f_\theta \mid \theta \in (\mathcal{S} \cap \mathcal{S}_t)\}$ is trained with a mini-batch of $D_m$ to preserve forward transfer and knowledge overlap between the past and the current tasks. The optimization learning rate is also considerably reduced at this stage compared to the *Retain* stage to prevent drastic weight changes in subnetworks, which in turn decreases the forgetting. This could be seen as an adaptation of the metaplasticity in the brain [26, 20] which refers to the ability to adjust the amount of plasticity in the network based on the current and future needs of the organism. At the end of this finetuning, the currently extracted subnetwork $\mathcal{S}_t$ is merged with the cumulative extracted mask from past tasks, $\mathcal{S}$. The merging of the subnetworks can be seen as a process of integrating new neurons ($\mathcal{S}_t$) into the existing neural network ($\mathcal{S}$), similar to the way neurogenesis allows for the integration of new neurons into existing neural circuits to accommodate new memories.

## 3.3 Rewind

Evidence implies that the brain uses active forgetting as a tool for learning due to its limited capacity [47, 29]. Studies also suggest that although learning and memory become more difficult to access during the forgetting process, they still exist [54, 48]. There are still memory remnants in the brain that can be reactivated [31]. In this work, we define the rewinding of weights to an earlier state as active forgetting. We also make sure that rather than reinitializing the model back to random or very

---

**Algorithm 1** Proposed Approach - TriRE

---

**input:** Data streams $\mathcal{D}_t$, working model $\Phi_\theta = g_\theta(f_\theta(.))$, EMA model $\Phi_{\theta_{EMA}} = g_{\theta_{EMA}}(f_{\theta_{EMA}}(.))$, sparsity factor $\gamma$, learning rates $\eta \gg \eta'$, *Retain* epochs $E_1$, *Revise* epochs $E_2$, *Rewind* epochs $E_3$.

1: $\mathcal{S} \leftarrow \{\}, \mathcal{M} \leftarrow \{\}$
2: **for all** tasks $t \in \{1, 2, .., T\}$ **do**
3:     **for** epochs $e_1 \in \{1, 2, ...E_1\}$ **do**                              ▷ *Retain*
4:         **for** minibatch $\{(x_i, y_i)\}_{i=1}^B \in \mathcal{D}_t$ and $\{(x_m, y_m)\}_{m=1}^B \in \mathcal{M}$ **do**
5:             Update $\{f_\theta \mid \theta \notin \mathcal{S}\}, g_\theta$ with $\eta$ on $\{(x_i, y_i)\}_{i=1}^B$ using Eq. 1 ($\mathcal{L}_t$)
6:             Update $\{f_\theta \mid \theta \in \mathcal{S}\}, g_\theta$ with $\eta$ on $\{(x_m, y_m)\}_{m=1}^B$ using Eqs. 1 ($\mathcal{L}_{er}$) and 2
7:         **if** $e == k$ **then**
8:             Save the weights $\theta_k$
9:         Update $\theta_{EMA}$ using Eq. 3
10:     Extract new subnetwork $\mathcal{S}_t$ with $\gamma$ sparsity based on CWI from Eq. 4
11:     **for** epochs $e_2 \in \{1, 2, ...E_2\}$ **do**                            ▷ *Revise*
12:         **for** minibatch $\{(x_i, y_i)\}_{i=1}^B \in \mathcal{D}_t$ and $\{(x_m, y_m)\}_{m=1}^B \in \mathcal{M}$ **do**
13:             Finetune $\{f_\theta \mid \theta \notin (\mathcal{S} \cap \mathcal{S}_t)\}, g_\theta$ with $\eta'$ on $\{(x_i, y_i)\}_{i=1}^B$
14:             Finetune $\{f_\theta \mid \theta \in (\mathcal{S} \cap \mathcal{S}_t)\}, g_\theta$ with $\eta'$ on $\{(x_m, y_m)\}_{m=1}^B$
15:             Update $\theta_{EMA}$
16:     Update cumulative set $\mathcal{S} = \mathcal{S} \cup \mathcal{S}_t$
17:     Reinitialize non-cumulative weights $\{f_\theta \mid \theta \notin \mathcal{S}\}$ with $\theta_k$
18:     **for** epochs $e_3 \in \{1, 2, ...E_3\}$ **do**                            ▷ *Rewind*
19:         **for** minibatch $\{(x_i, y_i)\}_{i=1}^B \in \mathcal{D}_t$ **do**
20:             Update $\{f_\theta \mid \theta \notin \mathcal{S}\}, g_\theta$ with $\eta$ on $\{(x_i, y_i)\}_{i=1}^B$
21:             Update $\theta_{EMA}$
        Update buffer $\mathcal{M}$
22: **return** model $\Phi_\theta$, model $\Phi_{EMA}$

---

early weights, it is rewound to a point where the model has learned some features and has a generic perception of the objective closer to convergence (but not absolute convergence). To aid this, the weights from a later epoch $k$ from the *Retain* phase are saved to be used in the *Rewind* stage.

Specifically, after the *Retain* and *Revise* steps, we rewind the weights belonging to non-cumulative subnetwork $\{f_\theta \mid \theta \notin \mathcal{S}\}$ back to the epoch $k$ weights. Then the rewinded weights are finetuned for a few epochs using a mini-batch of $D_t$. This is helpful because studies [46, 13] show that in the human brain, less active neurons follow a *'use-it-or-lose-it'* philosophy. Therefore, forgetting and relearning act as a warm-up for these less active neurons making them relevant again for the learning circuit and making them more receptive to learning the next task.

In summary, TriRE (Retatin-Revise-Rewind) involves iteratively applying the three phases mentioned above to each task within a lifelong learning setting. Our method effectively combines multiple biological phenomena and harnesses the advantageous characteristics provided by popular CL approaches. The step-by-step procedure is given in Algorithm 1.

## 4 Results

**Experimental Setup:** We expand the Mammoth CL repository in PyTorch [9]. On the basis of Class-IL and Task-IL scenarios, we assess the existing CL techniques against the proposed one. Although the training procedure for Class-IL and Task-IL is the same, during inference, Task-IL has access to the task-id. We consider a number of rehearsal-based, weight regularization, and parameter-isolation approaches as useful baselines because TriRE necessitates experience rehearsal and model modularity. We use ResNet-18 [21] as the feature extractor for all of our investigations. In order to reduce catastrophic forgetting, we additionally offer a lower bound SGD without any support and an upper bound Joint where the CL model is trained using the full dataset.

**Experimental Results:** We compare TriRE with contemporary rehearsal-based and weight regularization methods in Class-IL and Task-IL settings. As shown in Table 1, TriRE consistently outperforms

Table 1: Comparison of prior methods across various CL scenarios. We provide the average top-1 (%) accuracy of all tasks after training. † Results of the single EMA model.

| Buffer size | Methods | Seq-CIFAR10 | | Seq-CIFAR100 | | Seq-TinyImageNet | |
|---|---|---|---|---|---|---|---|
| | | Class-IL | Task-IL | Class-IL | Task-IL | Class-IL | Task-IL |
| - | SGD | $19.62_{\pm0.05}$ | $61.02_{\pm3.33}$ | $17.49_{\pm0.28}$ | $40.46_{\pm0.99}$ | $07.92_{\pm0.26}$ | $18.31_{\pm0.68}$ |
| | Joint | $92.20_{\pm0.15}$ | $98.31_{\pm0.12}$ | $70.56_{\pm0.28}$ | $86.19_{\pm0.43}$ | $59.99_{\pm0.19}$ | $82.04_{\pm0.10}$ |
| - | LwF | $19.61_{\pm0.05}$ | $63.29_{\pm2.35}$ | $18.47_{\pm0.14}$ | $26.45_{\pm0.22}$ | $8.46_{\pm0.22}$ | $15.85_{\pm0.58}$ |
| | oEWC | $19.49_{\pm0.12}$ | $68.29_{\pm3.92}$ | - | - | $7.58_{\pm0.10}$ | $19.20_{\pm0.31}$ |
| | SI | $19.48_{\pm0.17}$ | $68.05_{\pm5.91}$ | - | - | $6.58_{\pm0.31}$ | $36.32_{\pm0.13}$ |
| 200 | ER | $44.79_{\pm1.86}$ | $91.19_{\pm0.94}$ | $21.40_{\pm0.22}$ | $61.36_{\pm0.35}$ | $8.57_{\pm0.04}$ | $38.17_{\pm2.00}$ |
| | DER++ | $64.88_{\pm1.17}$ | $91.92_{\pm0.60}$ | $29.60_{\pm1.14}$ | $62.49_{\pm1.02}$ | $10.96_{\pm1.17}$ | $40.87_{\pm1.16}$ |
| | CLS-ER† | $61.88_{\pm2.43}$ | $\mathbf{93.59}_{\pm0.87}$ | $43.38_{\pm1.06}$ | $\mathbf{72.01}_{\pm0.97}$ | $17.68_{\pm1.65}$ | $52.60_{\pm1.56}$ |
| | ER-ACE | $62.08_{\pm1.44}$ | $92.20_{\pm0.57}$ | $35.17_{\pm1.17}$ | $63.09_{\pm1.23}$ | $11.25_{\pm0.54}$ | $44.17_{\pm1.02}$ |
| | Co$^2$L | $65.57_{\pm1.37}$ | $93.43_{\pm0.78}$ | $31.90_{\pm0.38}$ | $55.02_{\pm0.36}$ | $13.88_{\pm0.40}$ | $42.37_{\pm0.74}$ |
| | GCR | $64.84_{\pm1.63}$ | $90.8_{\pm1.05}$ | $33.69_{\pm1.40}$ | $64.24_{\pm0.83}$ | $13.05_{\pm0.91}$ | $42.11_{\pm1.01}$ |
| | DRI | $65.16_{\pm1.13}$ | $92.87_{\pm0.71}$ | - | - | $17.58_{\pm1.24}$ | $44.28_{\pm1.37}$ |
| | TriRE | $\mathbf{68.17}_{\pm0.33}$ | $92.45_{\pm0.18}$ | $\mathbf{43.91}_{\pm0.18}$ | $71.66_{\pm0.44}$ | $\mathbf{20.14}_{\pm0.19}$ | $\mathbf{55.95}_{\pm0.78}$ |

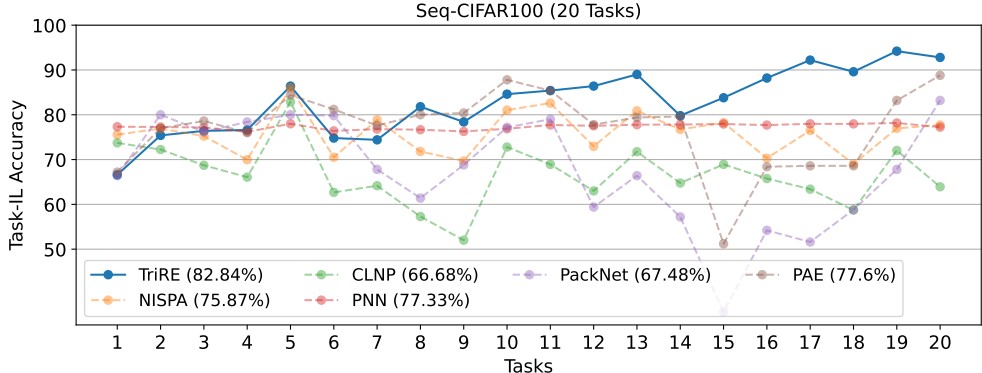

Figure 3: Comparison of TriRE against evolving architectures in terms of Task-IL accuracy on Seq-CIFAR100 dataset divided into 20 tasks. The graph reports the accuracy of individual tasks at the end of CL training.

rehearsal-based methods across most datasets, highlighting the significance of dynamic masking in CL. Although methods like Co$^2$L and ER-ACE excel on simpler datasets, they struggle with more challenging ones. The same applies to methods like DRI and GCR, which augment memory buffers through core-set and generative replay. Their performance, for instance, lags behind TriRE in Seq-TinyImageNet, where the buffer-to-class ratio is low. Retaining task-wise dynamic subnetworks and revising extracted subnetworks to preserve past knowledge significantly reduces task interference in TriRE. In essence, TriRE boosts generalization through weight and function space regularization, selective forgetting, and relearning, yielding superior performance across tasks.

As evident from Table 1, weight regularization methods such as LWF, oEWC, and SI perform miserably in both Class-IL and Task-IL settings across datasets. The reason being, these approaches encounter classes solely from the current task at any point in CL training. Therefore, they fail to discriminate between classes from different classes resulting in subpar performance. On the other hand, TriRE leverages samples from previous classes through experience rehearsal to learn discriminatory features across tasks. In addition, TriRE entails forming subnetworks through *Retain* and *Revise* resulting in modularity and reduced task interference between tasks.

Parameter isolation approaches minimize task interference in CL by creating distinct sub-networks either within a given model capacity or by dynamically growing the network. Figure 3 illustrates a comparison between methods trained on Seq-CIFAR100 with 20 tasks i.e., it depicts final accuracies

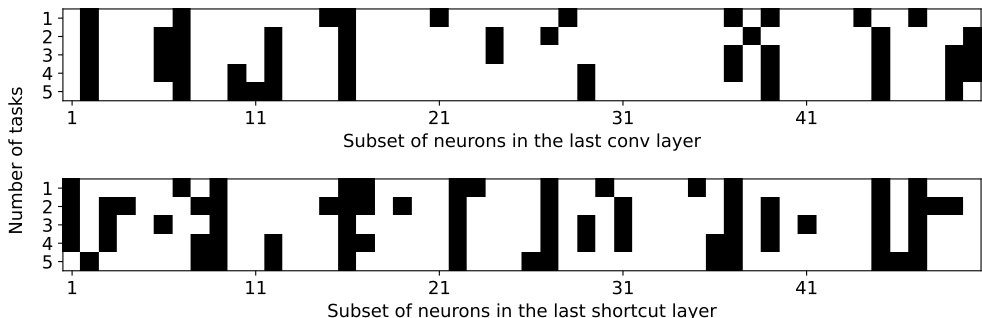

Figure 4: (Top) depicts the distribution of neuron activations across tasks in the last convolutional layer of the feature extractor and (Bottom) depicts the same for the last shortcut layer. The black cubes represent the extracted ones after *Retain* stage.

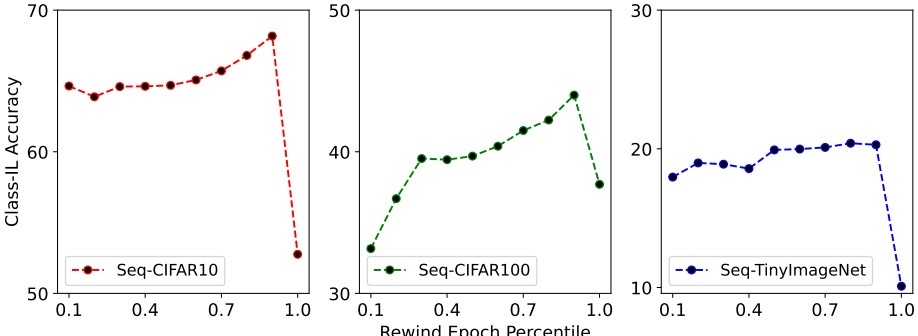

Figure 5: The effect of rewinding on Class-IL accuracy for all three datasets. The region from 70% to 90% of all epochs gives the best results consistently across datasets.

on $1^{st}$, $2^{nd}$.. $20^{th}$ task after training. Upon completing all tasks, TriRE achieves an average accuracy of 80.85%, surpassing the performance of the baselines considered. TriRE leverages the benefits of experience rehearsal, weight regularization, and function space regularization to learn compact and overlapping subnetworks, resulting in reduced task interference while maintaining scalability. In line with biological principles, TriRE incorporates selective forgetting and relearning mechanisms to activate less active neurons and enhance their receptiveness to learning subsequent tasks, thereby mitigating the risk of capacity saturation.

## 5   Model Analysis

**Task Interference:** Figure 4 shows the changes in neuronal activity for the subset of neurons in the last convolutional layer (top) and the last shortcut layer (bottom) of ResNet-18 trained on 5 tasks in Seq-CIFAR100 dataset. The neurons that are blacked out are the most active neurons for each particular task after the *Retain* phase. For each task, it is observed that there are exclusive subnetworks forming on their own (horizontally) that capture task-specific information. However, with CL training, several neurons become generalizable by capturing information that can be shared across tasks. Therefore, there is neuron overlap between the extracted neurons across tasks (vertically). More so in the shortcut layer, since the number of parameters in these layers is low. TriRE optimally manages the model capacity vs. task modularity trade-off better by re-using neurons that can be shared across tasks while maintaining the modularity of knowledge in each task intact.

**How Much to Rewind?** In Figure 5, we examine Class-IL accuracy when the model is rewound to different points in the *Retain* stage in order to comprehend how rewinding affects the accuracy of the model. We observe that the accuracy of the inference model decreases if the model forgets too much and is rewound to an early stage in the training. This is in alignment with the observations made by [14] that rewinding to extremely early stages is not recommended for DNNs because the network has

Table 2: Comparison of the contribution of each phase in TriRE. Note that the combination of *Revise* alone or *Revise & Rewind* has not been considered, as it is not feasible without the *Retain* phase.

| Retain | Revise | Rewind | Seq-CIFAR100 | | Seq-TinyImageNet | |
|---|---|---|---|---|---|---|
| | | | Class-IL | Task-IL | Class-IL | Task-IL |
| ✓ | ✗ | ✗ | 38.01 | 66.23 | 11.54 | 40.22 |
| ✓ | ✓ | ✗ | 33.08 | 60.03 | 8.44 | 31.90 |
| ✓ | ✗ | ✓ | 43.03 | **72.09** | 16.25 | 48.89 |
| ✓ | ✓ | ✓ | **43.91** | 71.66 | **20.14** | **55.95** |

Table 3: Relative number of learnable parameters and corresponding memory footprint in Seq-CIFAR100 with varying number of task sequence.

| Methods | Learnable Parameters (Million) | | | Memory Consumption (Million) | | |
|---|---|---|---|---|---|---|
| | 5 Tasks | 10 Tasks | 20 Tasks | 5 Tasks | 10 Tasks | 20 Tasks |
| DER ++ | 1x | 1x | 1x | 1x | 1x | 1x |
| EWC | 1x | 1x | 1x | 3x | 3x | 3x |
| TriRE | 1x | 1x | 1x | 6x | 6x | 6x |
| PNNs | 27x | 79x | 240x | 27x | 79x | 240x |

not learned enough meaningful features by then to regain the lost accuracy. Additionally, we notice that accuracy also suffers when the model is rewound to a point extremely close to the end of training time. Rewinding to a very late point in the training phase close to convergence is not ideal because there is not enough time for relearning. Our experiments indicate that rewinding to between 70% and 90% of the training time results in the best accuracy.

**Ablation Study:** As explained previously, TriRE employs a three-stage learning paradigm to reduce task interference and improve weight reuse in CL. We seek to uncover how each of *Retain*, *Revise*, and *Rewind* in TriRE influence Class-IL and Task-IL accuracies in Seq-CIFAR100 and Seq-TinyImageNet datasets through Table 2. It can be seen that although *Retain* alone can extract the subnetworks containing the most active neurons and decrease task interference, it falls short in aspects of forward transfer and weight reuse. Similarly, *Retain* and *Revise* together can solidify the knowledge extracted from current and past tasks, but such a model suffers from capacity issues without the reactivation of less active neurons for future tasks. Likewise, *Retain* and *Rewind* together can encourage task-wise delimitation of knowledge and promote efficient usage of available networks, but lose out on the forward transfer introduced by the learning of joint distributions. Finally, analogous to the brain, it is evident that the harmony of all components is what achieves the best results in both datasets.

**Memory and Computational Cost:** We conduct a comparative analysis of the computational and memory overhead of TriRE in contrast to related works. Table 3 provides an analysis of the learnable parameters and memory required by TriRE in contrast to those of DER++, EWC, and PNNs, (Opting for one method from each individual family of CL methods). Firstly, similar to DER++ and EWC, TRiRE does not add any learnable parameters to the model. However, it is evident that PNNs have an infeasible amount of learnable parameters which gets progressively worse with longer task sequences. Secondly, the observed increase in memory consumption in TriRE can be attributed to several factors: (1) the application of multiple masking mechanisms for parameter isolation; (2) the incorporation of the Rewind phase necessitating weight retention from a previous epoch; and (3) the utilization of the Exponential Moving Average (EMA) model to enhance knowledge consolidation. All of these factors hold memory but do not add any learnable parameter to the training.

## 6   Conclusion

We introduced TriRE, a novel biologically inspired CL paradigm that entails experience rehearsal, scalable neurogenesis, and selective forgetting and relearning to effectively mitigate catastrophic forgetting in CL. Within each task, TriRE entails retaining the most prominent neurons for each task, revising the extracted knowledge of current and past tasks, and actively promoting less active neurons for subsequent tasks through rewinding and relearning. Analogous to multiple neurophysiological mechanisms in the brain, TriRE leverages the advantages of different CL approaches, thus significantly lowering task interference and surpassing different CL approaches when considered in isolation. For Seq-TinyImageNet, TriRE outperforms the closest rival in rehearsal-based baselines by 14%,

surpasses the best parameter isolation baseline by 7%, and nearly doubles the performance of the best weight regularization method. Extending our method to CL scenarios oblivious to task boundaries and to few- and zero-shot learning settings are some of the future research directions for this work.

## 7 Limitations and Future Work

We proposed TriRE, a novel paradigm that leverages multiple orthogonal CL approaches to effectively reduce catastrophic forgetting in CL. As orthogonal CL approaches may not always be complementary, the selection of such approaches needs careful consideration in TriRE. In addition, having multiple objective functions naturally expands the number of hyperparameters, thereby requiring more tuning to achieve optimal performance. Therefore, additional computational complexity and memory overhead due to the staged approach and extensive hyperparameter tuning are some of the major limitations of the proposed method. For the same reason, we highlight that TriRE is not directed toward compute-intensive architectures such as vision transformers.

TriRE involves different stages of training within each task, requiring knowledge of task boundaries. In line with state-of-the-art methods in CL, each task entails a non-overlapping set of classes, and data within each task is shuffled to guarantee i.i.d. data. However, in the case of online learning where data streams and the distribution gradually shift, TriRE cannot be applied in its current form. Therefore, additional measures such as task-boundary approximation and modification to learning objectives are necessary to enable TriRE to work in such scenarios. Furthermore, traditional CL datasets considered in this work entail independent tasks and data points without intrinsic cumulative structure. As TriRE does not leverage structures learned in previously encountered tasks, structure learning forms another limitation of this proposed method. Reducing computational and memory overhead, extending to task-free CL scenarios with recurring classes, and leveraging intrinsic structures within underlying data are some of the future research directions for this work.

## 8 Broader Impacts

Inspired by the multifaceted learning mechanisms of the brain, we propose TriRE, which replicates the brain's ability to leverage multiple mechanisms for CL, enhancing the generalization capabilities of CL models across tasks. Its success not only encourages the exploration of neuro-inspired methods for deep neural networks, but also opens up opportunities to augment existing CL approaches by leveraging the advantages of competing strategies. By enabling models to learn continuously and adapt to new tasks, TriRE contributes to the responsible and ethical deployment of AI technologies, as models can improve and update their knowledge without requiring extensive retraining. This advancement has significant implications for various real-world applications and promotes the development of AI systems that can continually improve and adapt their performance.

**Acknowledgement:** The work was conducted while all the authors were affiliated with NavInfo Europe, Eindhoven, The Netherlands.

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

# A    Effect of Pruning Criteria

We seek to illustrate the effectiveness of different pruning criteria in TriRE. As explained in Section 3.1, the dense network is first pruned using k-WTA criteria, resulting in a subnetwork of the most activated neurons, and then this subnetwork is pruned using CWI criteria, resulting in a final extracted subnetwork at the end of *Retain* stage. Table 4 demonstrates the comparison of Class-IL accuracy between various pruning criteria, namely, magnitude-based, Fisher information-based, and CWI-based, across all three datasets. The idea behind magnitude pruning is that small valued weights impact the network's output less and can be safely pruned without significantly affecting performance. Fisher information-based pruning evaluates the importance of connections based on their contributions to the Fisher information matrix. Connections with low contributions, indicating less relevance or importance, are pruned or set to zero. However, both these criteria calculate the importance of weights within the current task, but do not consider the possibility of it being crucial for other tasks. On the other hand, CWI considers the significance of weights with respect to data saved in the rehearsal buffer as well, resulting in superior performance across all datasets.

Table 4: Comparison of the effect of various pruning criteria in TriRE on different datasets.

| Dataset | Magnitude | Fisher Information | CWI |
|---|---|---|---|
| Seq-CIFAR10 | $65.09_{\pm 0.83}$ | $64.40_{\pm 0.43}$ | $\mathbf{68.17}_{\pm 0.33}$ |
| Seq-CIFAR100 | $41.89_{\pm 0.74}$ | $40.26_{\pm 0.21}$ | $\mathbf{43.91}_{\pm 0.18}$ |
| Seq-TinyImageNet | $19.07_{\pm 0.97}$ | $18.16_{\pm 0.75}$ | $\mathbf{20.14}_{\pm 0.19}$ |

# B    Model Analysis

## B.1    Task Recency Bias

In any CL setting, the model entails learning on a few or no samples from previous tasks while aplenty of the most recent task [22]. This tilts learning toward the most recent task, resulting in decisions biased toward new classes and confusion among the old classes. However, the CL model should ideally have predictions distributed evenly across all tasks with the least possible recency bias. Figure 6 provides the confusion matrix for various CL models to evaluate the task recency bias. After training on Seq-CIFAR100 for 5 tasks with a buffer size of 200, the model is deemed to have correctly predicted the task label if it predicts any of the classes that make up the sample's true task label. As can be seen, ER and DER++ have a propensity to frequently classify the majority of samples as classes in the most recent task. However, TriRE's predictions are uniformly distributed across the diagonal. TriRE essentially decreases interference between tasks, captures task-specific information through extracted sub-networks, and produces the least recency bias.

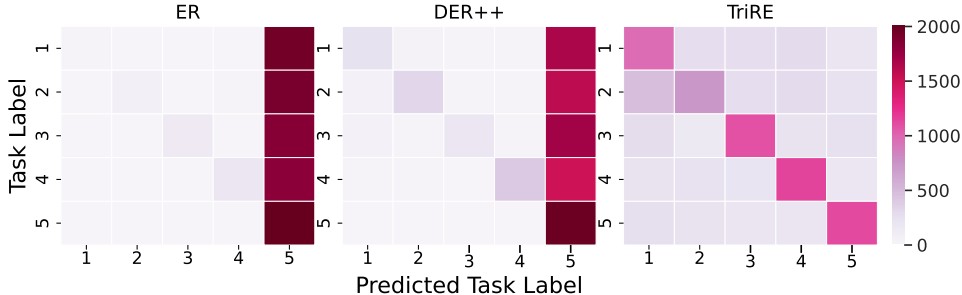

Figure 6: Confusion matrix of different rehearsal-based CL models. Unlike ER and DER++, TriRE predictions are evenly distributed across the tasks with the least recency bias.

## B.2    Stability-Plasticity Trade-off

A CL model is said to be stable if it can retain previously learned information, and plastic if it can effectively acquire new information. The stability-plasticity dilemma refers to an inherent trade-off

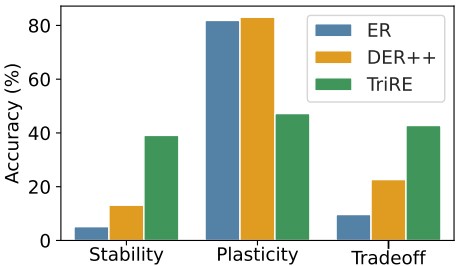 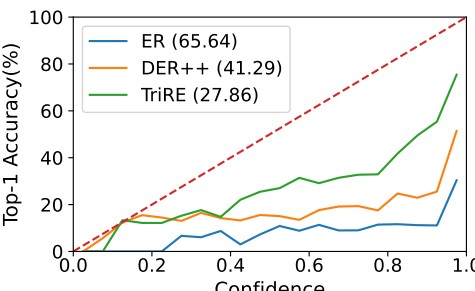

Figure 7: (Left) Stability-Plasticity Trade-off for CL models trained on Seq-CIFAR100 with 5 tasks. ER and DER++ are more plastic than stable leading to recency bias. TriRE maintains a better balance between stability and plasticity and achieves the highest trade-off amongst the baselines. (Right) Reliability diagram depicting model calibration: The red dashed line represents the ideal scenario. Compared to the other two methods, TriRE is better calibrated with the lowest ECE value. All models were trained on Seq-CIFAR100 with 5 tasks.

in which the CL model masters one of these aspects at the expense of the other. Sarfraz *et al.* [44] introduced a trade-off measure that serves as an approximation of how the model balances its stability and plasticity. Once the model completes the final task $T$, its stability ($S$) is assessed by calculating the average performance across all preceding $T - 1$ tasks as follows:

$$S = \sum_{i=0}^{T-1} A_{Ti} \tag{5}$$

The plasticity of the model (P) is evaluated by computing the average performance of each task after its initial learning i.e.,

$$P = \sum_{i=0}^{T} A_{ii} \tag{6}$$

Thus, the trade-off measure determines the optimal balance between the stability ($S$) and the plasticity ($P$) of the model. This measure is calculated as the harmonic mean of $S$ and $P$.

$$\textit{Trade-off} = \frac{2SP}{S + P} \tag{7}$$

Figure 7 (Left) provides the stability-plasticity trade-off measure for different CL methods across different datasets for a buffer size of 200. ER and DER++ exhibit high plasticity, enabling them to rapidly adapt to new information. However, they lack the ability to effectively retain previously acquired knowledge. On the other hand, TriRE exhibits substantially high stability with low plasticity, resulting in a higher stability-plasticity trade-off.

### B.3 Model Calibration

Ensuring the reliability of safety-critical CL systems necessitates the presence of a well-calibrated model. Calibration refers to the task of accurately predicting probability estimates that reflect the true likelihood of correctness. Miscalibration, on the other hand, refers to the disparity between confidence and accuracy expectations. To assess the degree of miscalibration in classification, the Expected Calibration Error (ECE) involves partitioning the predictions into bins of equal size and calculating the difference between the weighted average of accuracy and confidence within each bin. A lower ECE value indicates better calibration in the underlying models.

Figure 7 (Right) shows a comparison of different CL approaches using a calibration framework trained on Seq-CIFAR100 with a buffer size of 200. Well-calibrated CL systems accurately represent the true likelihood of accuracy (indicated by the red dashed line). Among the baselines, TriRE achieves the lowest ECE value and exhibits high calibration, demonstrating its effectiveness in minimizing task interference and reducing overconfidence in CL, thus enabling more informed decision making.

## C   Hyperparameter Selection

The hyperparameters required to replicate the results of TriRE can be found in Table 5. These hyperparameters were determined through a tuning process involving different random initializations and a small portion of the training set reserved for validation. All experiments were conducted using a batch size of 32 and trained for 50 epochs. TriRE was optimized using the Adam optimizer [24] implemented in PyTorch. Furthermore, the number of epochs allocated to each phase specified in Algorithm 1 was consistently set at a ratio of $E_1 : E_2 : E_3 = 3 : 1 : 1$.

Table 5: Best hyperparameters of TriRE chosen for optimal performance on different datasets.

| Dataset | $\eta$ | $\eta'$ | $\gamma$ | $\lambda$ | EMA Parameters | | Rewind |
| | | | | | $\mu$ | $\zeta$ | Percentile |
| --- | --- | --- | --- | --- | --- | --- | --- |
| Seq-CIFAR10 | 0.0006 | 0.0001 | 0.4 | 0.06 | 0.999 | 0.18 | 0.9 |
| Seq-CIFAR100 | 0.002 | 0.0001 | 0.2 | 0.04 | 0.999 | 0.12 | 0.9 |
| Seq-TinyImageNet | 0.002 | 0.0001 | 0.3 | 0.05 | 0.999 | 0.01 | 0.8 |

### C.1   Hyperparameters Sensitivity Analysis

In order to showcase the robustness of TriRE to a choice of values for each hyperparameter, we conducted additional experiments by finetuning different hyperparameters. Specifically, we evaluated the performance of TriRE in the Seq-CIFAR100 dataset with a buffer size of 200 and 5 tasks. The results are visualized in the Figure 8. Our experimentation focused on adjusting hyperparameters such as sparsity, k-winner sparsity, and EMA model update frequency. Upon examining the graphs, it is evident that performance remains relatively stable across a range of values for these hyperparameters. This observation suggests that satisfactory results can be achieved without an exhaustive hyperparameter search.

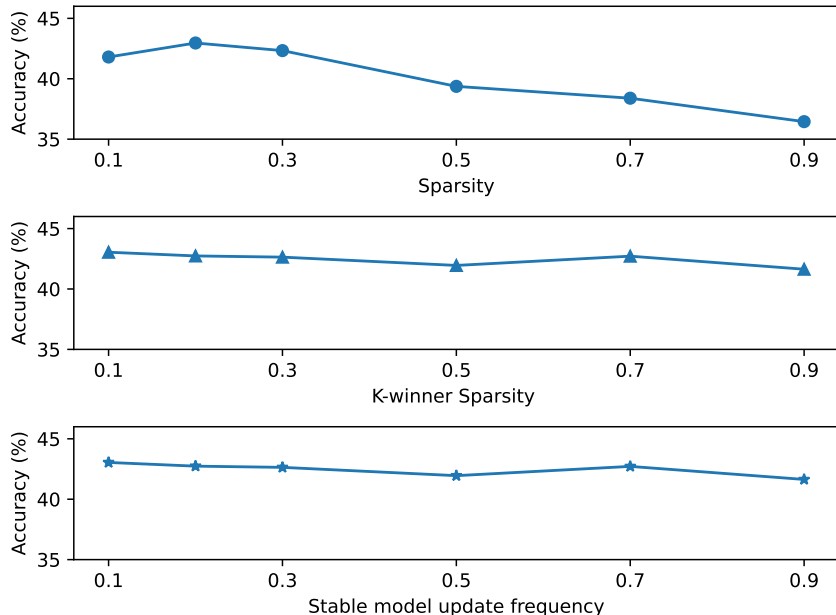

Figure 8: Evaluation of performance across different values of Sparsity, K-Winner Sparsity, and Stable model update frequency in Seq-CIFAR100 with buffer size 200. As can be seen, the performance does not drop significantly for different choices of values for hyperparameters.

# D    TriRE vs. CLS-ER: Commonalities and Differences

CLS-ER [6] emulates the interplay between fast and slow learning systems by incorporating two supplementary semantic memories that aggregate the weights of the working model in a stochastic manner via an exponential moving average. That is, CLS-ER operates with three models: the working model, the stable model, and the plastic model. Conversely, in the TriRE framework, the architecture consists of two models: the working model and a stable model (referred to as the EMA model). Therefore, the similarity is that both methods use the concept of progressively aggregating the weights of the working memory as it sequentially learns tasks allowing us to consolidate the information efficiently.

However, there are two main differences. Firstly, CLS-ER uses two supplementary semantic memories, whereas TriRE only utilizes one supplementary memory. Nevertheless, our results outperform CLS-ER (with one EMA model, referred to as Mean-ER in the mentioned paper) in all three datasets, as shown in Table 1. Secondly, CLS-ER only uses experience replay to tackle catastrophic forgetting whereas our method harmoniously integrates all the families of methods in CL, i.e. weight regularization, parameter isolation, and experience replay. That is, CLS-ER only focuses on one aspect of biological learning whereas we effectively consider multiple aspects.

# E    Comparison with Recent Baselines

Table 6 provides a comparison between Foster [51], Memo [57], and TriRE in Class-IL. The experiments are conducted on Seq-CIFAR100 with buffer size 200 with 5 tasks and 3 random seeds. We caution that the implementation details are slightly different: TriRE uses reservoir sampling with ResNet-18 as a backbone while both Foster and Memo employ modified ResNet-18 with custom sampling methods. Foster entails a dynamic expansion and compression to accommodate new information. Similarly, Memo entails expansion in the later layers while preserving generic information in the earlier layers. On the other hand, TriRE promotes generalization through a combination of weight and function space regularization, selective forgetting, and relearning thereby producing superior performance across tasks. As can be seen, TriRE outperforms both Foster and Memo.

Table 6: Comparison with Foster and Memo on Seq-CIFAR100 with 5 tasks and buffer size 200.

| Method | Foster | Memo | TriRE |
|---|---|---|---|
| Top-1 Accuracy (%) | 40.48 ±0.53 | 43.57 ±0.44 | **43.91** ±0.18 |

As the CL training progresses, both Foster and Memo are constrained to accommodate new information in a limited number of parameters resulting in lower performance in later tasks. However, TriRE suffers from no such limitation resulting in superior performance (see Figure 9).

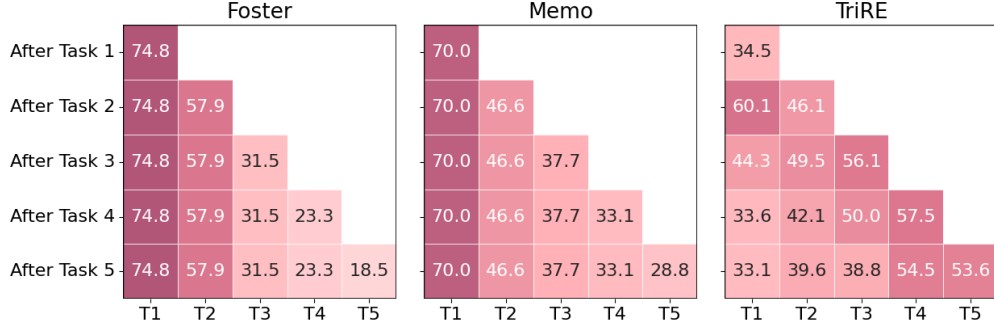

Figure 9: Task-wise performance on Seq-CIFAR100 with 5 tasks and buffer size 200. The models are assessed after completing each task (y-axis) to gauge how the progress of training impacts task performance (x-axis).

## F    Training Cost

We compare the training times of various CL models that were taken into account for this work. We consider one method from each family of CL models. Table 7 specifically lists the training times required to learn a single task (the first) for 3 epochs on an NVIDIA RTX 2080 Ti for Seq-CIFAR100 dataset with a buffer size of 200. We conducted the experiment for 3 epochs across all CL methods considering the three-phased approach in TriRE and to maintain the fairness of comparison.

Table 7: Training time comparison across various CL methods for three epochs on an NVIDIA RTX 2080 Ti for Seq-CIFAR100 dataset.

| Method | SI | DER++ | PNNs | TriRE | Phases of TriRE | | |
| --- | --- | --- | --- | --- | --- | --- | --- |
| | | | | | Retain | Revise | Rewind |
| Training Time (sec) | ∼32 | ∼72 | ∼25 | ∼75 | ∼33 | ∼30 | ∼12 |

Table 7 shows that PNNs have the least training cost for our experimental setup. However, it is to be noted that in the case of PNNs, as the number of tasks increases, the model size also increases. This indicates that the training time also eventually increases towards the later tasks. Furthermore, we see that SI also costs less training time-wise. However, according to Table 1, the Class-IL accuracies of weight regularization methods mentioned are significantly lower for larger datasets. Interestingly, DER++ and TriRE exhibit similar training costs, yet our approach proves to be more effective in mitigating catastrophic forgetting and task interference. This can be attributed to our method's unique amalgamation of various CL method families, making it a more robust choice for CL tasks.

We also compared the training costs for each stage within TriRE's learning paradigm. We found that 'Retain' incurs the highest cost among the three, as it involves activation and weight pruning. The 'Revise' stage, where the joint distribution of past tasks and the current task is learned, follows. Finally, 'Rewinding' to a saved weight and learning for a few epochs to activate less active neurons requires significantly less training time compared to the other two stages.

## G    Datasets and Settings

We assess the effectiveness of our approach in two different types of CL scenarios: Class Incremental Learning (Class-IL) and Task Incremental Learning (Task-IL). In Task-IL and Class-IL, each task consists of a predetermined number of new classes that the model needs to learn. A CL model learns multiple tasks in sequence while being able to differentiate between all classes it has encountered so far. Task-IL is similar to Class-IL, but it has the advantage of having access to task labels during the inference process, making it one of the easiest scenarios.

To evaluate the performance of our method in Task-IL and Class-IL scenarios, we employ three different datasets: Seq-CIFAR10, Seq-CIFAR100, and Seq-TinyImageNet. These datasets are derived from CIFAR10, CIFAR100, and TinyImageNet, respectively. In Seq-CIFAR10, CIFAR10 is divided into five tasks, each task containing two classes. Similarly, in Seq-CIFAR100, CIFAR100 is divided into five tasks, each consisting of 20 classes. Lastly, in Seq-TinyImageNet, we partition TinyImageNet into ten tasks, each of which comprises 20 classes. These datasets are designed to introduce more challenging scenarios for a comprehensive analysis of various CL methods. By increasing the number of tasks or the number of classes per task, we can thoroughly examine the effectiveness of different CL approaches in handling different levels of complexity. Following [6], we used ResNet-18 as the backbone in all our experiments. The training process remains consistent for both Class-IL and Task-IL. To compare various state-of-the-art approaches, we present the average accuracy across all tasks encountered in Class-IL. According to Task-IL conventions, we take advantage of task identity and selectively deactivate neurons in the linear classifier that are not related to the current task.

Contrary to the common practice of using dense CL models, dynamic sparse methods take a different approach by starting with a sparse network and maintaining the same level of connection density throughout the learning procedure to incorporate sparsity into a CL model; it is necessary to disentangle interfering units to prevent forgetting and establish new pathways to encode new knowledge. This presents challenges when implementing batch normalization and residual connections for both the NISPA and CLNP methods. Consequently, these methods do not employ the ResNet-18 architecture. Instead, they opt for a simpler CNN architecture without 'skip connections' and batch normalization.

