# OpenReview forum: "TriRE: A Multi-Mechanism Learning Paradigm for Continual Knowledge Retention and Promotion"
_NeurIPS.cc/2023/Conference — NeurIPS 2023 poster_

### Official Review · Reviewer_3Tuq · 2023-06-28

**Soundness:** 4 excellent
**Presentation:** 4 excellent
**Contribution:** 3 good
**Rating:** 8
**Confidence:** 4

**Summary:**

The paper concerns catastrophic forgetting in neural networks and draws from neural mechanisms contributing to the absence of such phenomena in the brain to propose a method to overcome CF through targeted neuron retraining, task knowledge revision, and enhanced learning for less active neurons. The authors present an impressive suite of experimental results, benchmarking their proposed method as superior or comparable to notable existing approaches in class- and task-IL scenarios and with further analysis of the different stages of the method.

**Strengths:**

+ The work creatively and effectively links to biological motivations toward continual learning (CL) capabilities.
+ Fairly inclusive summary of existing CL approaches with contributive discussion as to their strengths and weaknesses toward integration in the proposed approach.
+ Figures are well-crafted and easily understandable.
+ Performance is comparable to SOTA methods.
+ It's nice to see in the experimental data the significance of dynamic masking, retaining, rewinding.

**Weaknesses:**

- (As mentioned by the authors:) The work is primarily applicable only to CNN-based architectures in its present state, and additional tuning is required for the increased number of hyperparameters. While the paper is nice, one can see how TriRE won't work on transformers due to computational expense.

- Regarding Related Work, the authors could elaborate on whether there are any other notable CL approaches that draw upon strengths of previous works as TriRE does and clarify how TriRE is superior/comparable (if applicable).

- The authors could perhaps move discussion of limitations more heavily to the main text as opposed to the Appendix. The limitations discussed in Appendix C pertain to the practical applicability of the work and thus have merit for inclusion.

**Questions:**

* I am curious as to why rewinding improves accuracy, perhaps this could be another direction for exploration.

* Small clarifications regarding the method itself:
	* Learn: finite replay buffer with loss-aware experience rehearsal?
	* Retain: parameter isolation where important connections and weights are learned?
	* Revise: How are the subnetworks combined? I wonder if there is a way to specifically focus on free neurons when learning new tasks, so during the combination stage they are orthogonal to the combined network?
	* Rewind: Does relearning target specific unactivated neurons/parameters? This step is perhaps redundant if simply taking weights from a few epochs before and training more on new mini-batches to relearn.

Minor:
* Lines 159-160: Put “k” in math mode (top-k, k-winner)
* Line 215: “Retatin” -> “Retain”


**Limitations:**

The authors touch upon the limitations of their work in the main text and more so in the Appendix.

Social impact — not directly applicable but is mentioned in broader impacts discussion.

---

> ### Author Rebuttal · Authors · 2023-08-09
>
> We thank the reviewer taking time to review our work in detail. We appreciate your encouraging words on our manuscript. Our response to weaknesses and questions are as follows:
>
> `While the paper is nice, one can see how TriRE won't work on transformers due to computational expense.`
>
> We agree with the reviewer that added computational complexity is one of the major limitations of our approach. Therefore, a naive extrapolation of TriRE to transformer-based architectures might face some hurdles such as computational complexity, hyperparameter tuning etc. We extend our Limitations section to include these in the next revision. However, from a distinctive standpoint, TriRE is a novel training paradigm that enables effective assimilation of multiple orthogonal CL approaches and is one of the earliest works in this direction. Although TriRE suffers from some limitations, it has been successful in showcasing the combined efficacy of multiple orthogonal CL approaches. Also, it is to be noted that, other baseline methods mentioned in Table 1 and Figure 3 could also not work on transformers considering the architectures’ innate complexity and need for more resources and training time.
>
> `whether there are any other notable CL approaches that draw upon strengths of previous works as TriRE does`
>
> To the best of our knowledge, we are one of the earliest works in this direction. However, we might have missed some of the recent publications. As suggested by you and other reviewers, we intend to build upon our related works to provide more information on any other notable CL approaches that draw upon strengths of previous works as TriRE does.
>
> `move discussion of limitations more heavily to the main text as opposed to the Appendix`
>
> Due to the space constraints, we moved the Limitations section to Appendix. We intend to expand the limitation section further by incorporating suggestions from you and other reviewers for the revised version. Considering that, we would appreciate your guidance on identifying specific sections from the main paper that could potentially be moved to the appendix to make space for the Limitations section.
>
> `why rewinding improves accuracy, perhaps this could be another direction for exploration`
>
> There is existing literature which shows that active forgetting is a part of biological learning and that neuron decay due to unuse is one of the reasons for catastrophic forgetting. Inspired by that, we aimed the rewind and subsequent relearning to act as a warm-up for the less active neurons, that are not there in cumulative subnetwork S, making them relevant again for the learning circuit and engaging them to be more receptive to learning the next task. By keeping the cumulative subnetwork S intact, we also make sure that the rewinding of weights does not cause any performance drop. Moreover, we concur with the perspective that exploring the concept of "rewind" as a potential research direction holds promise and could open up new avenues for further investigation.
>
> `Learn: finite replay buffer with loss-aware experience rehearsal? `
>
> (Assuming the clarification is to understand loss-aware experience rehearsal better) In loss aware balanced reservoir sampling, we compute a score vector proportional to the number of items of each class and estimate an importance score given by the opposite of the loss value for each example. Then, we normalize these two terms to ensure an equal contribution and sum them to form a single score vector. Finally, we assign replacement probability to each item that is proportional to the combined score. This ensures that there is balance within the buffer in terms of the number of examples per class as well as help us identify and replace the elements displaying low loss values to make space for harder examples.
>
> `Retain: parameter isolation where important connections and weights are learned?`
>
> In the Retain phase, neurons and weights are learned for the current task. At the end of that stage, we employ k-WTA and heterogenous dropout based activation pruning and subsequent weight pruning to retain the best weights and activations for that task.
>
> `Revise: How are the subnetworks combined?`
>
> In the Revise stage, initially the cumulative subnetwork containing knowledge from past tasks and the subnetwork containing knowledge from the current task are trained / fine tuned together on the rehearsal buffer to learn their joint distribution. After this revising process, we update the cumulative set $S = S \cup S_t$. This makes sure that at the end of the current task, the cumulative set contains the best weights and activations of both past tasks and the current task which helps in preserving knowledge.
>
> `if there is a way to specifically focus on free neurons when learning new tasks, so during the combination stage they are orthogonal to the combined network?`
>
> We don’t encourage completely mutually exclusive subnetworks for each task as empirical analysis in Figure 4 shows that there is neuron overlap while learning different tasks. This knowledge sharing is particularly beneficial when it comes to scaling the model for longer task sequences. Furthermore, when all subnetworks are orthogonal to each other, it could lead to capacity saturation and task discovery problems during inference.
>
> `Rewind: Does relearning target specific unactivated neurons/parameters?`
>
> In the rewind phase, as correctly inferred by the reviewer, we are rewinding and relearning the less active neurons. The cumulative subnetwork S which contains the most important weights and activations remains unchanged. The rest of the less active neurons go through the Rewind phase.
>
> We hope that the provided clarification has addressed your concerns and inquiries to your satisfaction. However, if further assistance or elaboration is required, we would be more than happy to provide additional information to ensure a complete comprehension of our work.

---

> > ### Comment · Reviewer_3Tuq · 2023-08-14
> > **Thanks**
> >
> > Thank you for the detailed response.
> >
> > Regarding making space for limitations: I am not sure what would be feasible to remove while preserving the intended message and impact of each section -- however, the initial descriptions of each of the three stages (Section 3) could be made more concise to create more space overall; the Broader Impacts section could be shortened (last 2 sentences) and/or combined with the conclusion; and perhaps Algorithm 1 could be moved to the appendix, although I am not sure this is in line with best practice.

---

> > > ### Author Response · Authors · 2023-08-14
> > > **Reply to Reviewer 3Tuq**
> > >
> > > Thank you for your swift response.
> > >
> > > In line with the reviwer's suggestion, we will be moving the Limitations section to the main paper. Furthermore, we have expanded the scope of limitations to encompass the potential challenges that could arise from a naive extrapolation of TriRE to transformer-based architectures, notably with regards to computational complexity and memory overhead (Refer our 'revision plan' in the official comment for Reviewer tURo).

---

### Official Review · Reviewer_6DDf · 2023-07-03

**Soundness:** 2 fair
**Presentation:** 3 good
**Contribution:** 2 fair
**Rating:** 4
**Confidence:** 3

**Summary:**

The paper proposed a new continual learning (CL) method that updates partial neurons while rewinding other neurons to the previously stored weights. To do this, the authors utilize sparsity constraints to the network weights and select highly activated neurons. Consequently, the proposed method outperforms the baselines.


**Strengths:**

1. The proposed method is generally technically sound.
2. The ablation study in Table 2 helps us understand the contribution of each component.


**Weaknesses:**

1. The proposed method has three components, retain, revise, and rewind. The key contribution part is rewind while other parts are periphery or have no novelty.

2. What is the overhead cost for rewinding, such as training time,  memory cost for holding previous weights, and the performance drop for the current task?

3. There are 7 different parameters and it seems that it is arbitrarily chosen to make the best performance for each task (Appendix D) as the authors mentioned in Appendix C, which is appreciated.

4. The statement in L211-212, “This is helpful because studies show that in the human brain, less active neurons follow a ‘use-it-or-lose-it’ philosophy” is difficult to understand why it is “helpful.” It seems that the authors blindly think that “human-like” is good.

5. It will be better if the authors include recent baselines [1-2] and check whether the proposed method is state-of-the-art in the current setting.

[1] Fu-Yun, et al. "Foster: Feature boosting and compression for class-incremental learning." ECCV. 2022.

[2] Zhou, Da-Wei, et al. "A model or 603 exemplars: Towards memory-efficient class-incremental learning." ICLR. 2023


**Questions:**

See Weakness

**Limitations:**

See Weakness

---

> ### Author Rebuttal · Authors · 2023-08-09
>
> We would like to express our sincere gratitude for your thorough review of our paper. Below, we have addressed each of your concerns to the best of our understanding, aiming to enhance the paper's overall contribution.
>
> `What is the overhead cost for rewinding, such as training time, memory cost for holding previous weights, and the performance drop for the current task? `
>
> We appreciate the reviewer's insightful observation of the absence of computation cost analysis. Furthermore, we concur with the interpretation regarding the potential drawback of the Rewind phase in terms of memory consumption and training time. Yet, we wish to emphasize that the novelty in our paper lies in effectively integrating existing neuro-inspired concepts, aiming to guide the CL research community towards a less-explored direction. We are actively considering the efficient interpretation and implementation of TriRE as a natural progression of this work. However, we acknowledge the present version's drawbacks, including memory and training costs, which will be duly addressed in the limitation section.
>
> Additionally, we would like to address the concern regarding the possible performance drop in the current task because of the Rewind phase. The analysis for this can be found in Section B.2 in the Appendix. We analyze this from the perspective of the stability-plasticity dilemma. As shown in Figure 7, while other baselines like ER and DER++ suffer from recency bias and are more plastic, TriRE clearly balances the stability - plasticity tradeoff managing to preserve existing knowledge from past tasks while acquiring new knowledge from current tasks. This is because while rewinding the weights, we only rewind the weights that are not in the cumulative subnetwork S. And to reiterate, by the time we reach the Rewind stage, S contains the most important weights and activations for the past and current task. So by keeping them intact and only rewinding the rest of the weights, we preserve what we have already learned and manage to bring the less active neurons back into the learning circuit.
>
> `It seems that the authors blindly think that “human-like” is good.`
>
> There is existing literature [1,2] which shows that active forgetting is a part of biological learning and that neuron decay due to unuse is one of the reasons for catastrophic forgetting. Therefore, we aimed the rewind and subsequent relearning to act as a warm-up for the less active neurons, that are not there in cumulative subnetwork S, making them relevant again for the learning circuit and engaging them to be more receptive to learning the next task. This is to also make sure that the model is scalable and is able to utilize the available parameters efficiently. Moreover, we prove this hypothesis with empirical analysis in Section 5 (under Ablation Study) which separately focuses on the impact of the Rewind phase on the overall algorithm.
>
> [1]	Shors, Tracey J., Megan L. Anderson, D. M. Curlik Ii, and M. S. Nokia. "Use it or lose it: how neurogenesis keeps the brain fit for learning." Behavioural brain research 227, no. 2 (2012): 450-458.
>
> [2]	Almond, N. M. "The Use-It-Or-Lose-It Theory; The Cognitive Reserve Hypothesis and the Use-Dependency Theory: Methodological Issues, Previous Research, Current Research and Future Perspectives." K, Edison.(Ed). Episodic Memory: Formation, Clinical Disorders and Role of Aging. Nova Science Publishers, Inc: New York (2014).
>
> `It will be better if the authors include recent baselines [1-2]`
>
> We extend our gratitude to the reviewer for bringing to our attention the possibility of establishing new baselines. That being said, the experimental setup of the suggested papers and TriRE are different so it will be difficult to incorporate that in the current version. However, your suggestion has enriched our perspective and we will extend our Related Works section to include these newer methods in the final version. Currently, we have made an earnest effort to consider baselines that are representative of each category of the existing CL methods that combat task interference and catastrophic forgetting.
>
> Once again, we thank you for your thoughtful evaluation and consideration. We have diligently tried to address each concern you raised, and we are committed to ensuring that our paper contributes meaningfully to the conference proceedings.

---

> > ### Comment · Reviewer_6DDf · 2023-08-12
> >
> > Thank the authors that respond to my concerns. I have follow-up questions.
> >
> > `Related to Overhead of rewind`
> >
> > I recommend measuring or computing the complexity of the overhead cost in terms of memory and runtime.
> >
> > `Related to `It will be better if the authors include recent baselines [1-2]`
> >
> > It would be better if the authors specify how the experimental setting is different which hinders the fair comparison. Both are not quite different settings from DER++ used in Table 1

---

> > > ### Author Response · Authors · 2023-08-13
> > > **Reply to Reviewer 6DDf**
> > >
> > > We thank the reviewer for taking the time to review our rebuttal. With regard to measuring computational and memory overhead, we plan to include this in the final revision (See our 'revision plan' in the official comment for Reviewer tURo).  As these experiments need to be standardized (with experimental settings, software environments, and underlying hardware) to make sure we are comparing apples to apples, we expect these experiments to take some time. We hope to provide these results before the discussion period ends, but unfortunately, time and limited computational capacity are proving to be a constraint. In any case, we will include these results in our final revision.
> > >
> > > We considered the most important baselines from different CL approaches to evaluate the efficacy of TriRE. We intend to compare and contrast Foster [1] and Memo [2] analytically in our final revision. Some of the key differences hindering our progress are (1) Different backbones (ResNet-18 in TriRE Vs ResNet-32 variant in Foster for CIFAR100 experiments), different buffer sampling strategies, different buffer sizes (500 Vs 2000) and different training schedules. The same arguments hold for Memo as well as it has similar experimental settings as Foster. With sufficient time, these differences can be addressed and methods can be compared on a common ground. As per the reviewer's suggestion, we will include an experimental evaluation entailing a comparison between these methods in the final revision. Our revision plan can be found in the official comment for Reviewer tURo.
> > >
> > > [1] Fu-Yun, et al. "Foster: Feature boosting and compression for class-incremental learning." ECCV. 2022.
> > >
> > > [2] Zhou, Da-Wei, et al. "A model or 603 exemplars: Towards memory-efficient class-incremental learning." ICLR. 2023

---

> > > > ### Author Response · Authors · 2023-08-15
> > > > **Requesting feedback from Reviewer 6DDf**
> > > >
> > > > As per your suggestions, we are working on experiments concerning computational and memory overhead, and the comparison between Foster and Memo. As the discussion period is drawing close, we kindly request your feedback. Please let us know if any concerns remain. We would be more than happy to provide additional information to ensure complete comprehension of our work. If otherwise, we kindly request that you adjust your score to reflect the improved confidence in our paper.

---

> > > > > ### Author Response · Authors · 2023-08-16
> > > > > **Comparison with Foster [1] and Memo [2] as requested by Reviewer 6DDf**
> > > > >
> > > > > Based on reviewer's suggestions, we provide a comparison between Foster [1], Memo [2] and TriRE in Class-IL. The experiments are conducted on Seq-CIFAR100 with buffer size 200 with 5 tasks and 3 random seeds. We caution that  the implementation details are slightly different  for these methods and defer the details to the the final revision.
> > > > >
> > > > > | Foster | Memo | TriRE |
> > > > > | ------------- | ------------- | ------------- |
> > > > > |  40.48 ± 0.53  | 43.57 ± 0.44  | __43.91 ± 0.18__  |
> > > > >
> > > > > Foster entails a dynamic expansion and compression to accommodate new information. Similarly, Memo entails expansion in the later layers while preserving generic information in the earlier layers. On the other hand, TriRE promotes generalization through a combination of weight and function space regularization, selective forgetting, and relearning thereby producing superior performance across tasks. As can be seen, TriRE outperforms both Foster and Memo.
> > > > >
> > > > > As the CL training progresses, both Foster and Memo  are constrained to accommodate new information in limited number of parameters resulting in lower performance in later tasks. However, TriRE suffers from no such limitation resulting in superior performance.
> > > > > The task wise performance can be found in the following anonymous link:
> > > > > [taskwise_performance](https://osf.io/pyfxw/?view_only=db15d0cbc14644febfa43068f4fa992a)
> > > > >
> > > > > We will update the above results with a detailed discussion and the figure in our final revision.
> > > > >
> > > > > [1] Fu-Yun, et al. "Foster: Feature boosting and compression for class-incremental learning." ECCV. 2022.
> > > > >
> > > > > [2] Zhou, Da-Wei, et al. "A model or 603 exemplars: Towards memory-efficient class-incremental learning." ICLR. 2023

---

> > > > > ### Author Response · Authors · 2023-08-17
> > > > > **Regarding computational and memory overhead as requested by 6DDf**
> > > > >
> > > > > We conduct a comparative analysis of the learnable parameters and memory required by TriRE in contrast to those of DER++, EWC and PNNs, (i.e each individual family of CL methods). Firstly, similar to DER++ and EWC, TRiRE does not add any learnable parameters to the model. However, it is evident that PNNs have an infeasible amount of learnable parameters which gets progressively worse with longer task sequences. Secondly, the observed increase in memory consumption in TriRE can be attributed to several factors: (1) the application of multiple masking mechanism for parameter isolation, (2) the incorporation of the Rewind phase necessitating weight retention from a previous epoch, and (3) the utilization of the Exponential Moving Average (EMA) model to enhance knowledge consolidation. All of these factors hold memory but does not add any learnable parameter to the training.
> > > > >
> > > > > It is also noteworthy that, the employed masking mechanism presents potential for further optimization. Future iterations could explore techniques like spatial hashing and compression to enhance the computational efficiency of TriRE. Additionally, TriRE exhibits scalability for longer task sequences, distinguishing it from PNNs. As illustrated in the accompanying table, the memory consumption in PNNs experiences exponential growth with the number of tasks, whereas TriRE maintains a fixed capacity model. This is because our work, through the Revise and Rewind phases, compels the model to learn the joint task distribution while also facilitating the involvement of less active neurons in the learning process throughout training.
> > > > >
> > > > > | Methods |  | Learnable  Parameters  (Million) |  |  | Memory  Consumption  (Million) |  |
> > > > > |---|---|:---:|---|---|:---:|---|
> > > > > |  | **5 Tasks** | **10 Tasks** | **20 Tasks** | **5 Tasks** | **10 Tasks** | **20 Tasks** |
> > > > > | DER ++ | 1x | 1x | 1x | 1x | 1x | 1x |
> > > > > | EWC | 1x | 1x | 1x | 3x | 3x | 3x |
> > > > > | TriRE | 1x | 1x | 1x | 6x | 6x | 6x |
> > > > > | PNNs | 27x | 79x | 240x | 27x | 79x | 240x |
> > > > >
> > > > > In summation, TriRE occupies an intermediary position among the aforementioned methodologies with regards to the learnables parameters and memory overhead. TriRE is as good as DER++ and EWC in terms of learnable parameters required but does significantly better than PNNs. It offers superior performance with increased memory overhead over DER++ and EWC, all the while establishing a novel precedent for the harmonious integration of various CL techniques and without capacity saturation as in PNNs.
> > > > >
> > > > > We will include this analysis in the final version of the paper. Please let us know in case we have missed something.

---

> > > > > > ### Author Response · Authors · 2023-08-20
> > > > > > **Reply to Reviewer 6DDf**
> > > > > >
> > > > > > As the discussion period is nearing its end soon, we eagerly await the reviewer's feedback on our revised plan. We've made significant strides in our revisions: for instance, we conducted comparisons with Foster and Memo as the reviewer recommended. Additionally, we incorporated a memory overhead analysis to offer readers a comprehensive perspective on our method. Currently, we are delving into other experimental areas such as computational overhead and robustness concerning hyperparameter selection, among others. Given these enhancements, we would appreciate it if the reviewer could indicate any lingering concerns.
> > > > > >
> > > > > > We have made every effort to address your concerns and sincerely hope for your continued engagement. If our revisions align with your expectations, we respectfully request a reconsideration of the score.

---

### Official Review · Reviewer_SoXi · 2023-07-05

**Soundness:** 2 fair
**Presentation:** 2 fair
**Contribution:** 2 fair
**Rating:** 5
**Confidence:** 4

**Summary:**

The paper proposes a method for avoiding catastrophic forgetting in continual supervised learning that operates in three stages. In the first stage (retain), a subnetwork for the current task is identified by detecting most/least activated neurons of the main network. In the second stage (revise), both the main network and the extracted subnetwork are re-trained / fine-tuned with examples from the current task and examples from a replay memory. The subnetwork is later integrated into the main network. In the final stage (rewind), the weights belonging to non-cumulative subnetworks are finetuned for a few epochs. The paper presents experimental results on three benchmark continual learning datasets, in both the task-incremental and class-incremental settings, by also comparing with counterpart methods in the area.

--Rebuttal--
I read the rebuttal, along with other reviewers' comments, and increased my score accordingly (to borderline accept) during the rebuttal phase.

**Strengths:**

- The originality of the paper relies on a new multi-stage method to tackle catastrophic forgetting which relies on the important concepts of modularity and example replay.

- The paper is in general easy to follow.

- The paper is properly aligned with literature in the area.

**Weaknesses:**

- Although the proposed approach touches on a lot of important points in catastrophic forgetting and continual learning, it seems to be a mere combination of existing ideas into a so-called three-stage approach, therefore undermining the novelty of the proposed mechanism. Furthermore, what seem to be novel aspects of the proposed approach are not explained in full detail, and therefore it is difficult to estimate their impact. For instance, in the retain phase, activation pruning is performed by using the existing heterogeneous dropout, while weight pruning used the existing CWI approach. For activation pruning, in lines 158-159, it is stated that a counter is used to determine the top k-winner activations. However, not technical details are provided regarding how this counter works. In the revise stage, lines 191-193 mention that the learning rate is "considerably" reduced at this phase; however, no insights onto how to decide what the reduction rate to use. For this same stage, in lines 195-199 it is mentioned that the S and S_{t} subnetworks are eventually merged; however, no technical details of how this merge occurs are provided. Finally for the rewind stage, in lines 209-214, it is unclear how to decide for how many epochs (k?) to rewind the network, what are the criteria, the requirements to decide this.

- The experiments lack analysis of computational cost (e.g. memory consumption, training time). As can be inferred from Algorithm 1 and section 3, the multi-stage procedure proposed in this paper involves several calculations and passes through the network/subnetwork. What is the cost of this multi-stage procedure? How does this compare to counterpart methods?

- In the experiments, some results are left unexplained while some others have an ambiguous explanation. For example, why does the proposed method seem to underperform on Task-IL for two of the three datasets evaluated? The explanation for the "miserably" (please change this word) performance of methods such as LwF and SI provided in lines 240-246 looks ambiguous and incorrect, since these methods have been used previously in Task-IL.

- From the results in Figure 3, it can be implied that the proposed method actually improves performance as the number of tasks increases. Is that the case? Did you run multiple task orders? Are tasks 10 to 20 naturally easier?

**Questions:**

Please refer to questions listed in the "weaknesses" section.

**Limitations:**

Limitations in terms of the network architecture used are clearly stated in the paper.

---

> ### Author Rebuttal · Authors · 2023-08-09
>
> Firstly, we would like to express our gratitude for the time and attention you dedicated to reviewing our paper. Below, we have carefully addressed each of your concerns to the best of our knowledge to enhance the paper's overall contribution.
>
> `no technical details are provided regarding how the activation counter works.`
>
> For a given task, each neuron in a layer is monitored for its frequency of activations during training. In essence, each neuron is given an activation counter that increases when a neuron’s activation is among the top-k activations in its layer. In the Retain phase, the likelihood that a neuron would be dropped is inversely proportional to its activation counts. As a result, the model is not only encouraged to preserve the knowledge of the current task but also learn the new task utilizing neurons that were less active during earlier tasks. At the end of each task, we reset this counter to make sure that we track the neuron activity of every task separately.
>
> `the learning rate is "considerably" reduced at Revise phase; however, no insights into how to decide what reduction rate to use.`
>
> We understand that the indication to decrease the learning rate “considerably” in the Revise stage could be misleading for the reader and appreciate you pointing it out. Our notion from the suggestion was that depending on the optimizer used, model architecture and the complexity of the dataset involved, slowing down the learning in the Revision stage can help us by not drastically overwriting the existing weights while simultaneously reaping the benefits of joint distribution based training. We acknowledge that this is a hyperparameter that needs tuning but for the datasets mentioned in the paper, we have provided the learning rates that gave us the best results in the Appendix which could be a good starting point for anyone in the community who wants to develop on this idea. Also as a thumb of rule, from our experiments, we found that $\eta'$ should be approximately 1/20th of the initial learning rate $\eta$ for smaller datasets and 1/10th for larger datasets.
>
> `mentioned that the S and S_t subnetworks are eventually merged; however, no technical details of how this merge occurs are provided.`
>
> By merging we mean updating the cumulative set S = S $\cup$ S_t at the end of the revise step. Specifically, the new cumulative mask includes those weights that were already part of the cumulative mask and the new set of weights that are identified to be most active by S_t. This information is part of Algorithm 1, line 16. In any case, we will update Figure 1 and provide more clarity in the next revision.
>
> `it is unclear how to decide for how many epochs (k?) to rewind the network`
>
> This has been addressed in Section 5 under “How much to Rewind?”. As explained in the paper, rewinding to very early and very late stages in the training tends to decrease the accuracy. The former does not work because the network has not learned enough meaningful features by then to regain the lost accuracy and the latter does not work because there is not enough time for relearning. The empirical analysis as shown in Figure 5 proves that rewinding to between 70% and 90% of the training time in the Retain phase results in the best accuracy.
>
> `explanation for performance of methods such as LwF and SI provided in lines 240-246 looks ambiguous and incorrect`
>
> We regret that the explanation for the unsatisfactory performance of LwF and SI when compared to our work seems ambiguous. However, we do want to point out that our method does considerably better (Refer Table 1) than weight regularization methods like LwF and SI and we were simply trying to bring out the difference in the accuracy in both scenarios. There is a relative increase of ~40% between LwF and TriRE in terms of Task-IL accuracy in Seq-CIFAR10. Furthermore, TriRE’s Task-IL accuracy shows a relative increase of ~200% in Seq-TinyImageNet which is a harder dataset considering its low buffer-to-class ratio. This proves that our method which uses a combination of weight regularization, experience replay and parameter isolation is better than using weight regularization alone. However, under the reviewer’s suggestion, we would be removing the word “miserably” as it’s an unfairly harsh description and would be adding more explanation.
>
> `implied that the proposed method actually improves performance as the number of tasks increases. Is that the case? Did you run multiple task orders? Are tasks 10 to 20 naturally easier?`
>
> We would like to clarify that Figure 3 provides results on all 20 tasks after training on Seq-CIFAR100 with 20 task training i.e. Figure 3 depicts final accuracies on 1st, 2nd.. 20th task after training. Therefore, it cannot be inferred that moving from 10 tasks to 20 tasks is naturally easier. On the contrary,  catastrophic forgetting worsens as the number of tasks in a sequence increases, referred to as long-term catastrophic forgetting. The number of samples in the buffer representing each previous task drastically reduces in longer task sequences resulting in poor performance. With the Revise phase forcing joint distribution based training thus preserving forward transfer and with the Rewind phase forcing less active neurons to participate in the task learning, TriRE combats task interference better. We employ the same task order as the compared baselines to maintain fairness in comparison. Therefore, Figure 3 contains a single task order averaged over multiple random seeds.
>
> We once again thank the reviewer for detailed feedback. We have made an utmost effort to address all the concerns raised. Please let us know in case we have missed something.

---

> > ### Comment · Reviewer_tURo · 2023-08-14
> > **Given author responsiveness re limitations and other points, I now more strongly favor acceptance**
> >
> > If there is an official way to revise my rating of the ms, I hope someone will let me know.  Otherwise, I'll just say here that I'm hopeful the paper can be accepted and presented at least as a poster at the conference.

---

> > > ### Author Response · Authors · 2023-08-14
> > >
> > > Thank you for advocating for our paper. In your official review session, there is an "Edit" button that will enable you to modify the initial review and rating. Your support is greatly appreciated.

---

> > > > ### Comment · Reviewer_tURo · 2023-08-14
> > > > **Done**
> > > >
> > > > I have edited my initial review, noted the author's plan to address limitations in the 'limitations' section of the review, and I have revised my official rating upward with the rationale explained in the current test in the 'limitations' box.

---

> ### Comment · Reviewer_SoXi · 2023-08-19
> **Authors' rebuttal**
>
> Thanks to the authors for their responses to my questions and concerns. After reading those, along with other reviewers' comments and responses to those comments from the authors, I am happy to increase my score as long as the authors commit to include in their final version of their paper the explanations and results of missing elements, in particular computational cost of the method compared to others and details of hyperparameter tuning (considering the large number of hyperparameters that are needed).

---

> > ### Author Response · Authors · 2023-08-19
> > **Reply to reviewer Soxi**
> >
> > We thank the reviewer for their response. As suggested by you and other reviewers, we fully commit to including experiments on computational overhead, hyperparameter tuning, and explanation for missing elements. As can be seen in official comments for Reviewer 6DDf, we have made some progress with respect to suggested changes.
> > Please find our complete revision plan under an official comment for Reviewer tURo. Please let us know in case any of your concerns are missing in the final revision plan.

---

### Official Review · Reviewer_tURo · 2023-07-06

**Soundness:** 3 good
**Presentation:** 3 good
**Contribution:** 2 fair
**Rating:** 7
**Confidence:** 4

**Summary:**

This article introduces a multi-faceted approach to continual learning. combining features of many other approaches and relying on a three-phase training process, such that, as each new task in a sequence of tasks is encountered, subsets of weights in the network are 'retained', 'revised' or 'rewound' to values that maintain old knowledge while beginning to learn the new task, facilitate integration of weights important for the new task with those important for previous tasks, and maintain plasticity for future learning, respectively.  The authors find advantages for this approach relative to a large range of other approaches, especially with more complex task settings, and perform several ablations helping to clarify the roles of the three stages, along with a few other explorations.

**Strengths:**

The paper cites a wide range of relevant related work and situates its approach within the context established by other approaches.  The other approaches considered seem fairly extensively sampled.  I appreciated the consideration of the effects of the different ablations which show the importance of the 'rewind' phase as well as the stability plasticity analysis shown in the appendix.

**Weaknesses:**

The work seems thoughtful in relation to the relevant existing literature but it may be that the CL paradigm as explored in this project is in need of re-framing, if new breakthroughs are to be achieved.  While the advantages relative to many of the baselines seem clear, I felt that the additional complexity of the TriRE scheme made the gaps between it and some of the other approaches relatively difficult to get excited about.  Some of the features that seem relatively important, such as CWI, are direct importations from previous work, making it difficult to assess whether TriRE really advances our understanding, given its additional complexity compared to most other work.

To me the greatest weakness of the approach is one that appears to be widely shared across the CL literature:  This is the fact that this and all of the cited work rely on triggering complex meta-processes at the boundaries between tasks in ways that seem very distant from what might occur in biological networks on in naturalistic continual learning settings where task boundaries are not announced.

We need approaches that can address the continual learning problem when tasks shift more gradually and task boundaries are not available.  The approach also exploits features limited to the setting in which tasks are defined by the fact that they use completely non-overlapping sets of output neurons (i.e. distinct class labels).  For example the Loss on the new task items in Eq 4 only considers the class labels relevant to the current task.  Finally, continual learning as explored in this literature is restricted to tasks with no intrinsically cumulative structure.  Catastrophic forgetting also occurs in settings where new learning could productively build on structure learned in previously learned tasks (in the way that addition builds on counting, multiplication builds on addition, etc).  The whole body of work thus seems narrowly focused on paradigms of limited general interest.

**Questions:**

I had difficulty understanding the CLS-ER results, which are almost as good as TriRE.  A clearer understanding of what the CLS-ER model shares with TriRE and how it differs from it would be useful in evaluating the contributions of TriRE over and about the use of an EMA model smoothing weights across tasks.  The note in Table 1 is insufficient for me to understand the relationship between CLS-ER and TriRE.

**Limitations:**

I initially wrote: "The paper mentions limitations in the appendix.  As stated they hint at some of the same limitations I see in the work as described above.  Per the importance placed on limitations in instructions to authors and reviewers, these ought to be placed in the main text."

The authors have addressed this concern through the discussion during the rebuttal, leading me to increase my rating to 'Accept'.  While I think the weaknesses described above still apply, they are, as I have said, largely shared by a mini-paradigm in which these issues have been addressed.  I hope the authors will be encouraged by acceptance of this work to work on a new paradigm.

---

> ### Author Rebuttal · Authors · 2023-08-09
>
> We sincerely appreciate the reviewer for providing thoughtful feedback and sharing a constructive evaluation of our work. Your valuable input has greatly contributed to the enhancement of our paper.
>
> `complexity of the TriRE scheme made the gaps between it and some of the other approaches relatively difficult to get excited about. `
>
> We agree with the implication that TriRE has additional complexity when compared to the other mentioned baseline methods. But the exciting element in what we are proposing is that there is a very plausible research direction which was under explored and has shown promise. Although the work before ours has pushed the boundary in CL domain, they have missed out on the benefits of effectively integrating several neuro-physiological processes emulating how biological learning works. We manage to coherently combine the concepts of meta-plasticity, active forgetting, relearning, neurogenesis and context gating to form a new CL paradigm and empirically prove that it is a viable direction with results.
>
> `the greatest weakness of the approach is one that appears to be widely shared across the CL literature:`
>
> We appreciate the reviewer for highlighting a common gap observed in both our proposed work and the related literature. Your discerning analysis has emphasized an area of significance that merits careful consideration and refinement. That being said, we would like to indulge in the argument that CL as a topic is in its early phase and what you suggested : task boundary less training, rehearsal free training; are all part of the desiderata of CL [1]. Therefore, as the domain is progressing, we believe that there is a lot of scope for improvement. For instance, neuro-symbolic continual learning [2] is a subtopic within CL that broadly deals with the reviewer’s suggestion; the concept of new learning productively building on structure learned in previously learned tasks.
>
> However, in this work, we would reiterate that our focus was on demonstrating the efficacy of harmoniously combining the existing CL methods to tackle catastrophic forgetting better and it achieves the same. Ours is one of the earliest works in this direction and we hope that it spurs the CL research community to explore more methods which combine neuro-physiological processes as in the biological brain.
>
> [1]	Farquhar, Sebastian, and Yarin Gal. "Towards robust evaluations of continual learning." arXiv preprint arXiv:1805.09733 (2018).
>
> [2]	Marconato, Emanuele, Gianpaolo Bontempo, Elisa Ficarra, Simone Calderara, Andrea Passerini, and Stefano Teso. "Neuro symbolic continual learning: Knowledge, reasoning shortcuts and concept rehearsal." arXiv preprint arXiv:2302.01242 (2023).
>
> `A clearer understanding of what the CLS-ER model shares with TriRE and how it differs from it would be useful`
>
> CLS-ER emulates the interplay between fast and slow learning systems by incorporating two supplementary semantic memories that aggregate the weights of the working model in a stochastic manner via an exponential moving average. That is, CLS-ER operates with three models: the working model, the stable model, and the plastic model. Conversely, in the TriRE framework, the architecture consists of two models: the working model and a stable model (referred to as the EMA model). Therefore, the similarity is that both the methods use the concept of progressively aggregating the weights of the working memory as it sequentially learns tasks allowing us to consolidate the information efﬁciently.
>
> However, there are two main differences:
>
> 1. CLS-ER uses two supplementary semantic memories whereas TriRE only uses one supplementary memory and still our results are better than CLS-ER in all three datasets as shown in Table 1 of the main paper.
>
> 2. CLS-ER only uses experience replay to tackle catastrophic forgetting whereas our method harmoniously integrates all the families of methods in CL, i.e weight regularization, parameter isolation and experience replay. That is, CLS-ER only focuses on one aspect of biological learning whereas we effectively consider multiple aspects.
>
> We once again thank the reviewer for detailed and insightful feedback. Please let us know in case we have missed any open points.

---

> ### Comment · Reviewer_tURo · 2023-08-12
> **What revisions do the authors plan?**
>
> I hope I'm not missing it but I didn't see signs of any intention to revise either in the overall rebuttal or in the response to my comments.   Before re-confirming my belief that the paper deserves consideration for acceptance, I'd like to see the authors present a revised ms with a limitations section included in the main paper, specifically addressing the concerns I have raised with the whole continual learning setup of using announced task boundaries to gate complex meta-processes.

---

> > ### Author Response · Authors · 2023-08-12
> > **Reply to Reviewer tURo**
> >
> > We thank the reviewer for swift response. We take reviewers’ feedback seriously and intend to accommodate all their concerns in the final revision. As uploading a revised manuscript is not a possibility due to NeurIPS guidelines and the rebuttal had a limited word count, we did not provide a revised version of the limitations section. Considering the feedback from all reviewers, we intend to update our manuscript with the following changes:
> >
> > - Computational and memory overhead comparison
> >
> > - Robustness to choice of hyperparameters
> >
> > - Information pertaining to differneces between CLS-ER and TriRE
> >
> > - Clarifications regarding Figure 3
> >
> > - Extending our evaluation with more recent baselines (as proposed by other reviewers)
> >
> > - Other miscellaneous / minor changes, clarifications
> >
> > - Limitations and Future Work:
> > We proposed TriRE, a novel CL paradigm that leverages multiple orthogonal CL approaches to effectively reduce catastrophic forgetting in CL. As orthogonal CL approaches may not always be complementary,  the selection of such approaches needs careful consideration in TriRE. In addition, having multiple objective functions naturally expands the number of hyperparameters, thereby requiring extensive tuning to achieve optimal performance. Therefore, additional computational complexity and memory overhead due to the staged approach and extensive hyperparameter tuning are one of the major limitations of the proposed method.  For the same reason, we highlight that TriRE is not directed towards compute-intensive architectures such as  vision transformers.
> > As TriRE involves different stages of training within each task, it assumes the knowledge of task boundaries. Moreover, in line with state-of-the-art methods in CL, each task entails a non-overlapping set of classes with data within each task shuffled to guarantee i.i.d. data. However, in the case of online learning where data is streaming and the distribution is shifting gradually, TriRE cannot be applied in its current form. Therefore, additional measures such as task-boundary approximation and modification to learning objectives are necessary to enable TriRE to work in such scenarios. Furthermore, traditional CL datasets considered in this work entail independent tasks and data points without  intrinsic cumulative structure. As TriRE does not leverage structures learned in previously encountered tasks, structure learning forms one other limitation of this proposed method. Reducing computational and memory overhead, extending to task-free CL scenarios with recurring classes, and leveraging intrinsic structures within underlying data are some of the future research directions for this work.

---

### Author Rebuttal · Authors · 2023-08-09

In this section, we aim to address the overarching concerns raised by reviewers, ensuring a comprehensive response to the broader themes highlighted in their feedback.

`The key contribution part is rewind while other parts are peripheral or have no novelty / It seems to be a mere combination of existing ideas into a so-called three-stage approach.`

We respectfully disagree with this characterization. We intend to showcase that the novelty of the proposed method lies in the fact that it is able to effectively combine the existing families of CL methods to form a sophisticated new CL paradigm. The key contribution of the work is not the stages or phases by themselves but our effective interpretation of how these stages can harmoniously work together to create a novel CL method. As mentioned in the ‘Related Works’ section, each of these categories of methods has a plethora of research available individually, but they also have their drawbacks. For instance, weight regularization (inspired by meta plasticity) alone only imposes a soft penalty, which doesn’t entirely prevent catastrophic forgetting. Rehearsal methods (inspired by experience replay) do better but suffer overfitting in low buffer regimes, whereas parameter isolation (inspired by context gating and/or neurogenesis) alone is not a good candidate for longer task sequences due to capacity saturation. Notably, mammalian brain doesn’t depend on a singular concept but on an amalgamation of the above mentioned neuro-physiological phenomena working coherently together. As obvious as it may seem, the current CL research is not focused in that direction, and with this work, we are trying to bring focus to this idea by showing its effectiveness and plausibility. To the best of our knowledge, ours is one of the earliest works in this direction and we have proved its potential with empirical results.

`Weight pruning uses the existing CWI criterion`

While we recognise that CWI is an existing weight pruning criterion, it is essential to note that the proposed work focuses on a novel CL paradigm and considers weight pruning as a means to an end. The purpose here is not to propose a new pruning method. Nonetheless, in the process, other pruning criteria were considered like magnitude pruning and fisher information based pruning (Refer file attached). As shown in the Table, it is evident that CWI only contributes marginally to the overall accuracy improvement. The method is conceptually compatible with other criteria as well. The rationale behind selecting CWI is primarily rooted in its capacity to assess the weight significance relative to data stored in the rehearsal buffer. As shown in the Table, even other pruning criteria yields better results than CLS-ER even without hyperparameter tuning due to time constraints. Also, as weight pruning is an active research area, we believe newer pruning criteria could also work in conjunction with our method.

`There are 7 different parameters and it seems that it is arbitrarily chosen`

As this work is one of the earliest ones in the direction of effectively combining existing CL methods, it is not the most optimal method and the number of hyperprarameters adds a perception of complexity. However, we can assure you that it is not arbitrarily chosen to make the best performance for each task. Although the combination of hyperparameters that gave the best accuracy have been tabulated in Table 4 as a means for reproducibility, our method is not highly sensitive to the particular choice of hyperparameters and is robust enough to handle different settings to attain similar performance. Regarding the intuition behind the hyperparameters, for instance, Rewind percentile, which decides how much to rewind, has been analyzed and the instinct for choosing values in the range of 70-90% percentile has been discussed in Section 5. Furthermore, the parameter is fixed at 0.999 for all experiments to mimic the slow acquisition of structured knowledge by the EMA model. Also, the learning rate in the Revise phase ($\eta$') is dependent on the initial learning rate ($\eta$). From our experiments, we found that for smaller datasets, $\eta$' is approximately 1/20th  of $\eta$ and for larger datasets, its 1/10th of $\eta$. It is a safe bet to use this criterion as a starting point to find the most optimal learning rate for the Revise stage.

`The experiments lack analysis of computational cost and what is the cost of multi-stage procedure?`

We appreciate the reviewers for identifying the lack of computation cost analysis, and we have made a note to include a comprehensive analysis in the final version of the paper. Also, we would agree with the extrapolation that the multi-stage procedure proposed in this paper is not the most optimal solution memory and computation efficiency-wise and has a lot of room for improvement. However, we would also like to reiterate that efficiency of the method was not a goal for this work and the novelty of this paper lies in the effective combining of existing  neuro-inspired concepts. We are trying to spur the CL research community towards this direction which is less explored. Efficient interpretation and implementation of TriRE is a natural progression to this work that we are also considering.

`Why does the proposed method seem to underperform on Task-IL for two of the three datasets evaluated?`

Considering Task-IL is considered to be an easier CL scenario than the Class-IL, TriRE’s Task-IL performance is not significantly worse or better statistically than the competing methods. However, as the CL scenario gets harder, the difference between the competing methods is clearer due to their ability to mitigate catastrophic forgetting. For instance, their performance is much worse compared to TriRE in Seq-TinyImageNet where the buffer-to-class ratio is low. We regret the lack of clarity in the current version and will update the manuscript as per your suggestion in the next revision.

---

> ### Comment · Reviewer_6DDf · 2023-08-12
> **Robustness to hyperparameters should be experimentally presented not verbally described.**
>
> The authors should show experimental results not verbally described. Otherwise, readers are difficult to believe that the proposed method is robust to hyperparameters although it requires many of them, like Figure 5.

---

> > ### Author Response · Authors · 2023-08-12
> > **Reply to Reviewer 6DDf**
> >
> > We thank the reviewer for their swift response. We appreciate the reviewer's feedback and acknowledge the importance of analysis depicting robustness to choice of hyperparameters. As mentioned earlier, Figure 5 is one such example that depicts the relationship between the choice of rewind percentile and TriRE performance. As per reviewer's suggestion, we will collate more such results with respect to other hyperparameters as well. Although we envisage to provide them before the end of the discussion period, we are constrained by time and limited computational capacity. In any case, we will report these results in the final revision.

---

> > > ### Author Response · Authors · 2023-08-21
> > > **More hyperparameter tuning experiments**
> > >
> > > In accordance with the recommendations of the reviewers, we have carried out additional hyperparameter tuning experiments and subsequently compared the performance of TriRE in the context of Seq-CIFAR100 with a buffer size of 200 and 5 tasks. The outcomes are detailed as graphs in the temporarily accessible anonymous link: [Tuning experiments link](https://osf.io/pyfxw/?view_only=db15d0cbc14644febfa43068f4fa992a)
> > >
> > > Our experimentation involved the manipulation of parameters such as sparsity, k-winner sparsity, and EMA model update frequency. As can be seen from the graphs, the performance does not vary significantly for a range of values for these hyperparameters.  This observation suggests that achieving satisfactory performance is achievable without the necessity for exhaustive hyperparameter tuning. We will include these results and a description in the final revision.

---

### Decision · Program_Chairs · 2023-09-21

**Decision:**

Accept (poster)

**Comment:**

This paper tackles class incremental learning from a new perspective of utilizing a multi-mechanism approach which includes rehearsal, network growing, forgetting, and re-learning. Results are shown across datasets such as sequential TinyImageNet including in low-buffer settings. Reviewers appreciated a number of aspects of the paper, including rethinking of the current paradigm where methods are studied in isolation, thoroughness of other approaches considered, and ablations showing the importance of the different phases. Weaknesses included complexity of the method (including computation and memory requirements), addition of significant hyper-parameters, triggering of various processes based on task boundaries, and comparison to the state-of-art. The authors provided a rigorous rebuttal with explanations and additional experiments: comparisons to Foster/Memo, computation/memory overhead, and hyper-parameter sensitivity.

  After considering these rebuttals, most reviewers expressed that their concerns have been largely satisfied and retained or increased to positive scores. One reviewer has retained a weak reject and expressed a number of concerns, but in my view many of them have been addressed satisfactorily. Overall, considering the paper, reviews, rebuttal, and discussions, I recommend acceptance of this paper and believe it adds an interesting multi-method perspective to the continual learning literature. I highly recommend that the authors incorporate the issues raised in the reviews and during discussions, for example limitations both of the method itself as well as the general continual learning settings used in the paper.